# OliTag-seq enhances in cellulo detection of CRISPR-Cas9 off-targets
Zhi-Xue Yang [1,2,4], Dong-Hao Deng [1,2,4], Zhu-Ying Gao [1,2,4], Zhi-Kang Zhang [3,4], Ya-Wen Fu [1], Wei Wen[1,2], Feng Zhang [1,2], Xiang Li[1,2], Hua-Yu Li [3] ✉, Jian-Ping Zhang [1,2] ✉ & Xiao-Bing Zhang [1,2] ✉

The potential for off-target mutations is a critical concern for the therapeutic application of CRISPR-Cas9 gene editing. Current detection methodologies, such as GUIDE-seq, exhibit limitations in oligonucleotide integration efficiency and sensitivity, which could hinder their utility in clinical settings. To address these issues, we introduce OliTag-seq, an in-cellulo assay specifically engineered to enhance the detection of off-target events. OliTag-seq employs a stable oligonucleotide for precise break tagging and an innovative triple-priming amplification strategy, significantly improving the scope and accuracy of off-target site identification. This method surpasses traditional assays by providing comprehensive coverage across various sgRNAs and genomic targets. Our research particularly highlights the superior sensitivity of induced pluripotent stem cells (iPSCs) in detecting off-target mutations, advocating for using patient-derived iPSCs for refined off-target analysis in therapeutic gene editing. Furthermore, we provide evidence that prolonged Cas9 expression and transient HDAC inhibitor treatments enhance the assay's ability to uncover off-target events. OliTag-seq merges the high sensitivity typical of in vitro assays with the practical application of cellular contexts. This approach significantly improves the safety and efficacy profiles of CRISPR-Cas9 interventions in research and clinical environments, positioning it as an essential tool for the precise assessment and refinement of genome editing applications.

The CRISPR-Cas9 system has catalyzed a paradigm shift in genome engineering, allowing for the meticulous alteration of DNA sequences with a degree of precision previously unattainable[1–4]. Central to this technology is the SpCas9 nuclease from Streptococcus pyogenes, which, in concert with a single guide RNA (sgRNA), orchestrates the induction of double-stranded breaks (DSBs) at predetermined genomic loci[5–8]. This capacity for targeted modification is the cornerstone of CRISPR-Cas9's utility. However, off-target mutations constitute a persistent challenge, casting a shadow on the system's reliability, particularly in clinical interventions where genomic accuracy is paramount[9–11]. The precise quantification and localization of these inadvertent edits are pivotal in understanding and mitigating the risks associated with CRISPR-Cas9, thereby bolstering its therapeutic applicability and safety profile.

A multitude of techniques has emerged to chart the off-target consequences of Cas9 activity, spanning cell-based systems like GUIDE-seq and iGUIDE to in vitro alternatives including Digenome-seq, CIRCLE-seq, and CHANGE-seq[12–16]. While in vitro methodologies exhibit heightened sensitivity, they often do not fully represent the chromatin intricacies inherent to living cells. This discrepancy can lead to the identification of numerous potential off-target sites that, in the end, may not be relevant or present within the cellular milieu[14,17]. Among in cellulo approaches, GUIDE-seq has gained traction for its use of a 34 bp double-stranded oligodeoxynucleotide (dsODN) to flag off-target interactions. Nonetheless, this method grapples with the challenge of inefficient oligo tag integration, a problem exacerbated by the tags' propensity for AT-rich sequences, resulting in less-than-ideal off-target event detection[15,18].

[1]State Key Laboratory of Experimental Hematology, National Clinical Research Center for Blood Diseases, Haihe Laboratory of Cell Ecosystem, Institute of Hematology & Blood Diseases Hospital, Chinese Academy of Medical Sciences & Peking Union Medical College, 300020 Tianjin, China. [2]Tianjin Institutes of Health Science, 301600 Tianjin, China. [3]College of Computer Science and Technology, China University of Petroleum (East China), 266000 Qingdao, China. [4]These authors contributed equally: Zhi-Xue Yang, Dong-Hao Deng, Zhu-Ying Gao, Zhi-Kang Zhang. ✉e-mail: lhyzj@upc.edu.cn; zhangjianping@ihcams.ac.cn; zhangxbhk@gmail.com

In this study, we introduce OliTag-seq, a refined version of GUIDE-seq, which utilizes a guanine-rich 39 bp dsODN designed to bolster both the stability and integration efficiency at the sites of Cas9-induced DSBs. By adopting a triple-priming technique in the PCR amplification phase, OliTag-seq substantially heightens the sensitivity for identifying off-target mutations. Empirical evidence from our investigations reveals that OliTag-seq exhibits superior performance over GUIDE-seq in pinpointing off-targets across a spectrum of sgRNAs and genomic contexts. Additionally, our research illuminates the particular aptitude of induced pluripotent stem cells (iPSCs) for off-target detection within the OliTag-seq framework, attributing this to their characteristically open chromatin structure that favors dsODN integration[19,20]. We further explore how the persistent expression of Cas9, coupled with the strategic application of histone deacetylase (HDAC) inhibitors, can amplify the assay's sensitivity to detect off-target events.

OliTag-seq facilitates the reliable detection of off-target sites and guarantees consistent reproducibility across technical and biological replicates. This method marks a notable advancement over current techniques by offering a sensitive, cost-efficient, and pragmatic solution for comprehensive off-target detection across the genome. Its deployment is poised to be particularly impactful in clinical research settings, where the precise mapping of off-target effects is critical for developing safe and effective gene therapies.

## Results

### Enhancing dsODN tagging efficiency for improved off-target detection

The original GUIDE-seq method utilized a blunt, 5'-phosphorylated 34 bp dsODN to label DSBs induced by Cas9, with dsODN integration and PCR amplification pivotal for off-target site identification (Fig. 1a). However, the AT-rich composition of this dsODN may have hindered complete integration at CRISPR cleavage sites, resulting in less efficient PCR priming. To address this, we modified the dsODN, extending it to 39 bp and integrating GC-rich clamps at both ends to enhance stability (Fig. 1b).

Our analysis at six genomic loci (*EEF2*, *BCL11a*, *ALB-In13*, *COL1A1*, *ITG2B*, and *GAPDH1*) showed that the 39 bp dsODN achieved a 1.4-fold increase in overall dsODN insertion compared to the 34 bp version (Supplementary Fig. 1a, b). We focused on full-length integration, as dsODN tag truncation could impede primer binding during PCR. The 39 bp dsODN demonstrated a 1.3 to 2.3-fold enhancement in full-length integration, suggesting that its stabilized ends may facilitate more effective end-joining with genomic DNA (Fig. 1c; Supplementary Fig. 1c). While both dsODN designs allowed tandem integration at Cas9-induced breaks, the efficiency was low (0–10%), producing 50 to 100 bp PCR byproducts, which were excluded via magnetic bead purification before secondary PCR (Supplementary Fig. 1d).

In exploring small molecules that enhance NHEJ-mediated dsODN integration, B02, a RAD51 recombinase inhibitor[21], and Mirin, targeting Mre11's nuclease activity, emerged as the most efficacious[22]. These inhibitors augmented the integration frequency by approximately 1.8-fold (Fig. 1d; Supplementary Fig. 1e). The action of Mre11 and B02, predominantly through the suppression of homologous recombination, appears to tip the balance towards favoring NHEJ, thereby facilitating dsODN integration. VE-822 exhibited moderate effects in this context[23], while RS1 and cyclosporin H showed minimal impact[24]. Consistent with its established role as an inhibitor, M3814 decreased dsODN integration, serving effectively as a negative control in these experiments[25].

To verify if the 39 bp dsODN's improved tagging translated to more efficient PCR amplification and off-target detection, we conducted electroporation experiments in K562 cells using a Cas9 plasmid, sgRNAs targeting *TRAC*, *EMX1*, and *VEGFA site 3* (*VEGF3*), along with dsODN templates. Sequencing data revealed a substantial enhancement, with the 39 bp dsODN registering a 2.6–11.7 fold increase in read counts (Fig. 1e; Supplementary Fig. 1f). This confirmed the 39 bp dsODN's improved efficiency. Notably, the 39 bp dsODN outperformed the 34 bp version in off-target detection (Fig. 1f; Supplementary Fig. 1g), solidifying its superior capabilities in identifying off-target events more conservatively and representatively.

In sum, these findings underscore that the 39 bp dsODN bolsters integration efficiency and facilitates more reliable full-length insertions, thereby significantly enhancing the sensitivity in detecting off-target sites.

### Improved off-target detection in OliTag-seq through triple-primer PCR amplification

In refining OliTag-seq, we implemented a triple priming strategy during the initial PCR phase, aiming to selectively amplify dsODN-gDNA junctions. This approach departs from the traditional GUIDE-seq protocol, which typically requires separate amplifications to sequence regions upstream and downstream of the dsODN insertion.

Our triple priming method introduces a distinctive primer for adaptor ligation, equipped with a Unique Molecular Identifier (UMI) to minimize amplification bias. The other two primers are designed to specifically target the dsODN integration points. We have carefully calibrated the primer ratios to 2:1:1 to achieve accurate and consistent amplification of the intended genomic regions (Fig. 2a). As a result, this primer configuration generates three distinct PCR products, encompassing both sequences adjacent to the tagged site and a full-length product covering the dsODN. Primer dimers and shorter fragments are effectively removed through size-based magnetic bead selection.

The subsequent PCR step utilizes barcoded primers that introduce sample-specific sequences at the 5' end of the forward primers. This adjustment simplifies the process and reduces costs for large-scale pooled sequencing, diverging from the GUIDE-seq protocol that integrates barcodes within the PCR primers and adaptors.

Our comparative assessment of the primers' efficiency demonstrated a notable increase in sequencing read counts, with an approximate 1.6-8.3 fold improvement using our mixed PCR method over the conventional dual PCR approach (Fig. 2b; Supplementary Fig. 2a). Notably, in the initial PCR, this innovative triple priming technique captured more off-target events (Fig. 2c; Supplementary Fig. 2b). This discovery highlights an improved sensitivity of OliTag-seq in detecting off-target modifications, suggesting that it may outperform the standard separate PCR method commonly used in GUIDE-seq.

### OliTag-seq: A Sensitive Approach for Detecting CRISPR-Cas9 Off-Target Sites in K562 and U2OS Cells

In evaluating the proficiency of OliTag-seq for identifying CRISPR-Cas9-induced off-target mutations, we employed the technique on K562 and U2OS cells, which had been transfected with Cas9 and sgRNA targeting the *VEGF3* locus. Seventy-two hours post-transfection, the top 20 predicted off-target sites were amplified using barcoded primers and subjected to next-generation sequencing to ascertain the presence of insertions and deletions (indels).

We analyzed editing frequencies and dsODN insertion efficiencies using CRISPResso2[26]. The indel frequencies at these off-target sites exhibited considerable variation, ranging from 0 to 80%, with dsODN insertion efficiencies spanning from 0 to 26%. We observed a strong positive correlation between dsODN insertion efficiencies and indel mutation frequencies ($R^2 = 0.85$, Fig. 3a). Furthermore, OliTag-seq read counts showed significant correlations with dsODN integration frequencies ($R^2 = 0.69$, Fig. 3b) and indel percentages ($R^2 = 0.67$, Supplementary Fig. 2c). These correlations suggest that OliTag-seq can serve as a reliable indirect indicator of cleavage efficiency at targeted sites.

The reproducibility of OliTag-seq was robust, as evidenced by the high correlation in sequencing read counts across technical replicates ($R^2 = 0.88$, Fig. 3c). Regarding biological reproducibility, consistently high-read off-target sites were verified across multiple biological samples. However, sites with read counts under 10 showed variable reproducibility, likely due to their low cleavage frequencies (Fig. 3d, e).

Our analysis employed stringent criteria for the classification of high-confidence off-target sites, necessitating their detection in at least two separate biological replicates. This method significantly bolsters the

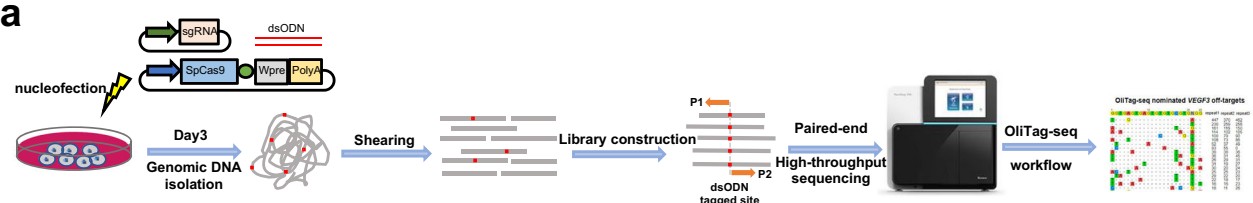

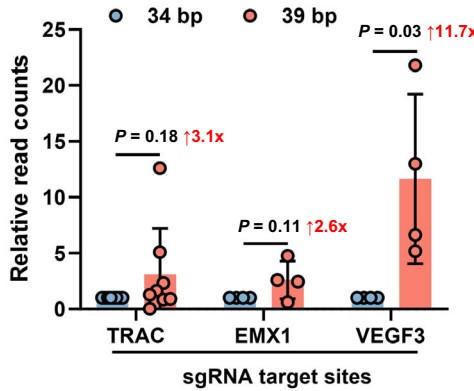

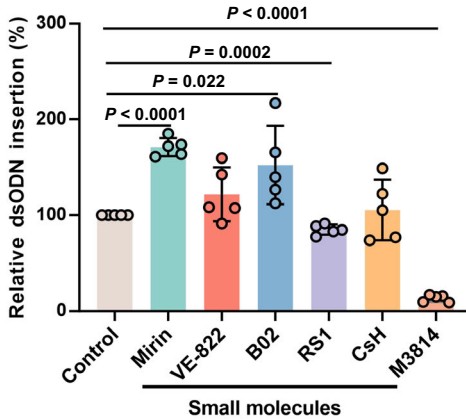

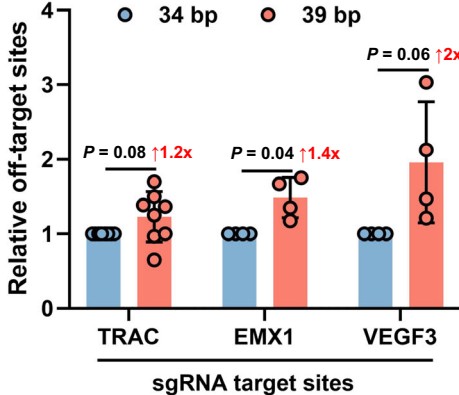

**Fig. 1 | Optimizing dsODN Tagging for Enhanced Detection of CRISPR-Cas9 Off-Target Effects. a** OliTag-seq Overview: The diagram details the OliTag-seq methodology for detecting CRISPR-Cas9 off-target effects. Cells are electroporated with Cas9-sgRNA and dsODNs, and genomic DNA is sequenced after three days per the OliTag-seq protocol, pinpointing dsODN-tagged DSBs. **b** dsODN Sequences Comparison: This section juxtaposes the dsODN sequences from GUIDE-seq (34 nt) with OliTag-seq (39 nt), outlining structural features like 5′ phosphorylation ('P') and phosphorothioate bonds ('*'). Extended GC clamps on the 39 nt ODNs are highlighted, indicating their potential for improved performance. **c** dsODN Integration Efficiency: The efficiency of full-length dsODN integration at six genomic sites is compared for 34 nt and 39 nt ODNs ($n = 3$–4 per site). The frequency of reads with complete ODNs relative to total edits is calculated, with the 39 nt ODN

efficiency normalized to the 34 nt ODN. **d** NHEJ-Mediated dsODN Integration and Small Molecules: This examines the effects of small molecules on NHEJ-mediated dsODN integration ($n = 5$), using M3814 as a negative control. Integration frequencies are normalized against controls to evaluate the potential enhancement by these treatments. **e** dsODN Tagging in K562 Cells: The analysis compares sequencing read counts for DSBs tagged with 34 nt and 39 nt dsODNs in K562 cells, following electroporation with Cas9 and specific sgRNAs. For balanced comparison, counts for 39 nt ODNs are normalized to 34 nt ODNs. **f** Off-Target Site Comparison: Off-target sites tagged by 39 nt and 34 nt ODNs in genes like TRAC, EMX1, and VEGF3 are compared ($n = 4$–8 per site). Normalized data for 39 nt ODNs are displayed as mean ± s.d., with statistical significance determined by paired two-tailed Student's $t$ tests and annotated with adjusted $p$-values.

reliability of our data. It is crucial to clarify that off-target sites identified in only one of three biological replicates are not dismissed as mere artifacts; instead, they are categorized as low-confidence off-targets, reflecting the occasional nature of such events.

## Superior sensitivity in off-target detection by OliTag-seq compared to GUIDE-seq

Our comparative analysis compared OliTag-seq against GUIDE-seq, applying it to U2OS cells using four sgRNAs targeting *EMX1*, *VEGF1*,

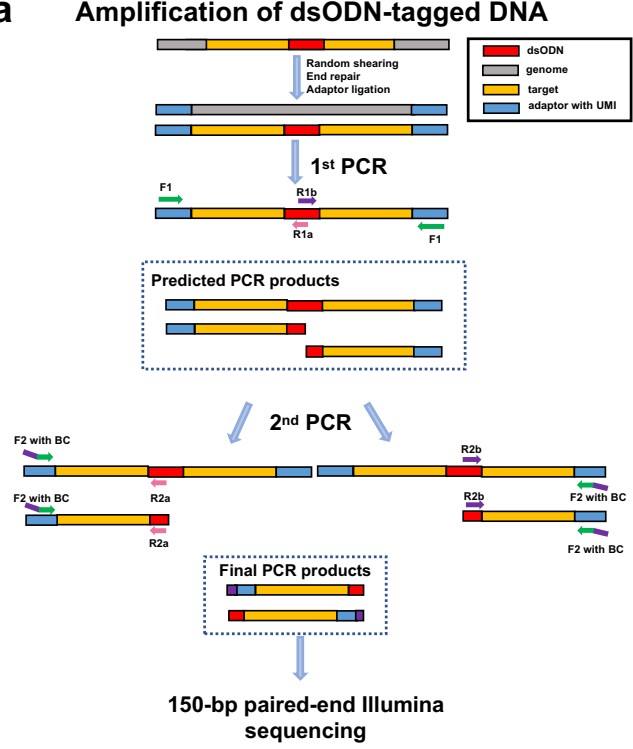

## a   Amplification of dsODN-tagged DNA

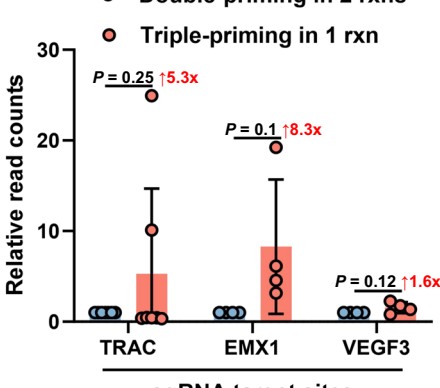

## b   Triple-priming enhances total counts

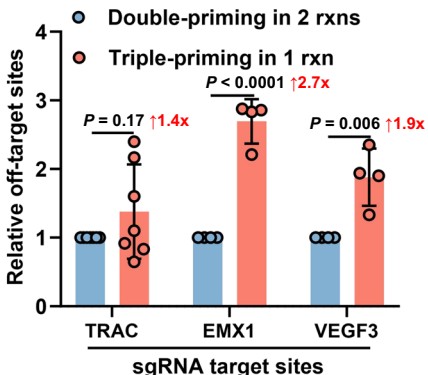

## c   Triple-priming amplifies off-target detection

**Fig. 2 | Optimized OliTag-seq Library Preparation via Triple Priming. a** Nested-PCR for gDNA Amplification: Genomic DNA, prepared with shearing and end-processing, is ligated to adaptors with unique molecular identifiers (UMIs). The initial PCR usesthree primers that anneal to the oligo tag and adaptor, creating a trio of product types. A second PCR stage amplifies these products to enrich flanking sequences around the dsODN insertion. The collective PCR output is then sequenced using 150 bp paired-end reads. **b** Enhanced Read Generation with Triple Priming: Triple priming in the first PCR markedly boosts read quantity for target sites compared to standard PCR methods, based on samples ($n$ = 4 or 7 per site). These triple-priming results are normalized against a baseline established by double priming across two reactions. **c** Improved Off-Target Detection via Triple Priming: A single-reaction triple priming approach yields more detectable off-target sites, with results standardized against a double-priming, two-reaction protocol. Data, sourced from sgRNA-directed targeting of TRAC, EMX1, and VEGF3, are expressed as means with standard deviation. Statistical significance is calculated using paired two-tailed Student's $t$ tests, with adjusted p-values detailed for clarity.

---

*VEGF2*, and *VEGF3*—genes previously mapped using GUIDE-seq. The results demonstrated OliTag-seq's heightened sensitivity, as it identified a more expansive array of off-target sites (ranging from 32 to 431 across the targets), encompassing all sites previously noted by GUIDE-seq for *VEGF1* (Fig. 4a–e and Supplementary Data 1). While OliTag-seq missed a few off-target sites detected by GUIDE-seq for the other sgRNAs, increasing sequencing depth could capture these additional sites.

Crucially, OliTag-seq unveiled a significantly larger number of off-target sites compared to GUIDE-seq for each tested sgRNA. Off-target sites identified by both methods tended to exhibit high read counts, indicating robust detection. In contrast, sites uniquely detected by OliTag-seq were characterized by lower read counts. This observation suggests that GUIDE-seq's lower dsODN integration efficiency might limit its detection range.

Furthermore, we closely examined around 28 newly identified off-target sites for *EMX1*, *VEGF1*, *VEGF2*, and *VEGF3* in gene-edited U2OS cells without the ODN template. Subsequent analyses confirmed mutations at these novel off-target sites identified by OliTag-seq, with validation through amplicon sequencing revealing indels (Supplementary Fig. 3a-d). These collective findings underscore OliTag-seq's superior capability in detecting CRISPR-Cas9 off-target sites, representing a noteworthy advancement over GUIDE-seq.

### iPSCs demonstrate enhanced sensitivity for off-target site detection compared to other cell lines

Our exploration of the sensitivity of different cell lines in off-target site detection extended the study beyond the commonly used U2OS cells in GUIDE-seq. This assessment encompassed K562, HEK293T, and iPSCs, utilizing sgRNAs targeting *EMX1*, *VEGF1*, and *VEGF3* (Supplementary Figs. 4–8 and Supplementary Data 2). Notably, U2OS cells showed the lowest dsODN insertion frequency and, correspondingly, identified the fewest off-target sites. K562 cells, despite a higher dsODN insertion frequency, detected a limited number of off-target sites (Fig. 5a, b).

Interestingly, HEK293T cells pinpointed more off-target sites than K562 cells, despite having a lower dsODN insertion efficiency. This discrepancy suggests that additional factors, beyond insertion frequency, play a role in off-target site detection (Fig. 5d–f). Remarkably, iPSCs excelled over all other tested cell lines in off-target identification for 2 out of 3 sgRNAs, uncovering over 100 off-target sites for *VEGF3*—almost double the count of other cell types (Fig. 5b, f).

We introduced an' off-index' metric to systematically evaluate off-target effects across these cell lines, where higher values denote a greater likelihood of off-target cleavage. This index is derived from the ratio of total off-target to on-target reads. U2OS and K562 cells registered the lowest off-index scores

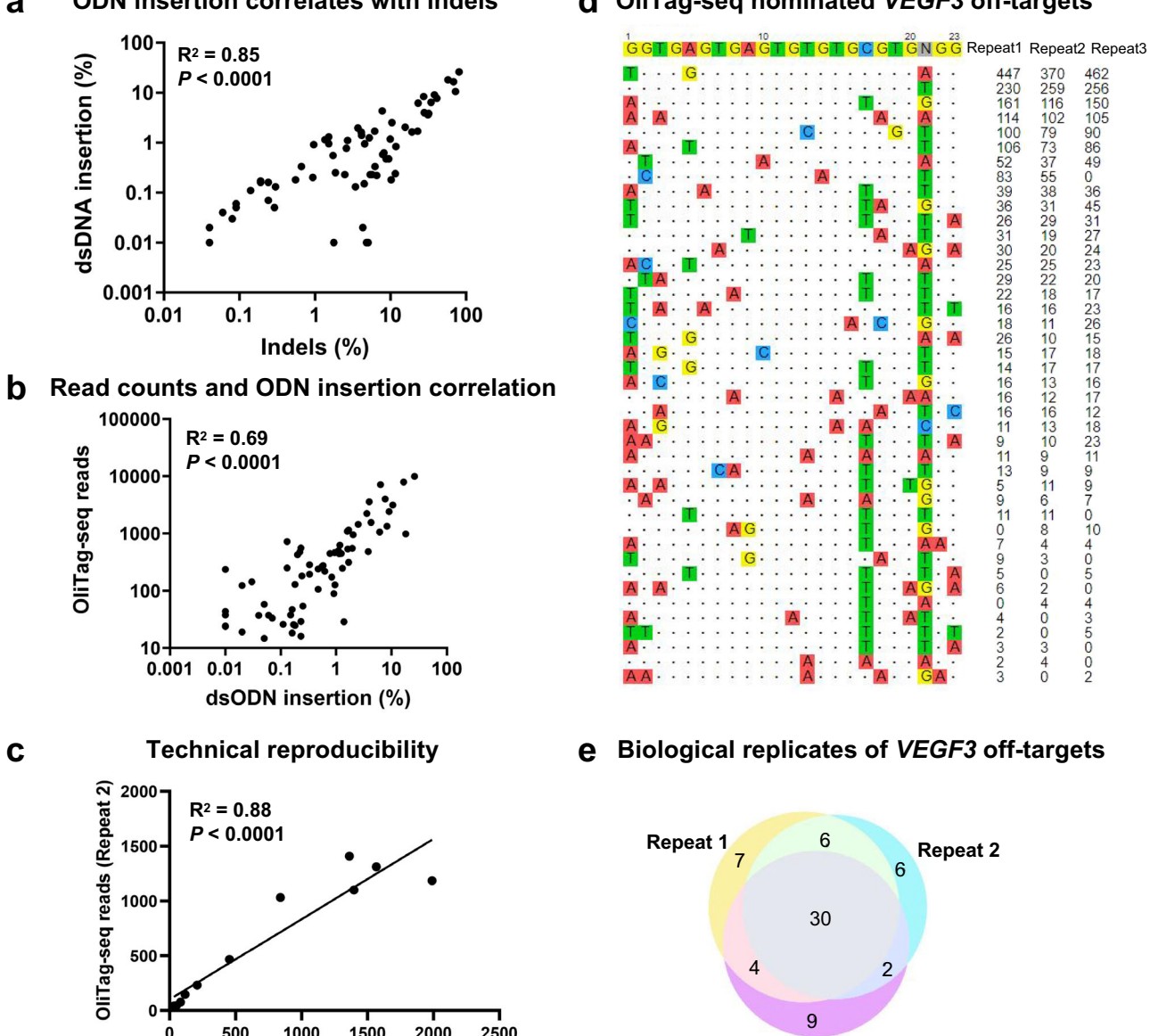

**Fig. 3 | Consistency in OliTag-seq Outcomes. a** Editing Frequency vs. dsODN Insertion Efficiency: Illustrates the relationship between total editing frequencies and dsODN insertion efficiencies, with the x-axis detailing editing occurrences and the y-axis showing dsODN integration rates. **b** dsODN Integration Frequency and Read Counts: Scatterplots map dsODN integration rates at on-target and off-target sites (x-axis) against corresponding sequencing reads (y-axis) ascertained by OliTag-seq for sgRNAs directed at VEGFA site 3. **c** Technical Replicate Correlation: Demonstrates the consistency between two technical replicates, assessing the reliability of the OliTag-seq method. **d** Off-Target Site Visualization for Replicates: Shows the reproducibility of off-target site detection across three biological replicates when targeting VEGFA site 3 in K562 cells. The reference on-target sequence with its PAM is prominent at the top, with mismatches in off-target sequences accentuated through color coding. OliTag-seq read quantities are denoted alongside. **e** Replicate Overlap Analysis: A Venn diagram depicts the commonality among three biological replicates for sgRNA targeting VEGFA site 3 in K562 cells, highlighting the reproducibility of identified off-target sites. Pearson linear regression analysis substantiates the data in panels A-C, with p-values for Pearson's r calculated using a two-tailed t-distribution, validating the method's precision and reproducibility.

in line with the identified off-target sites. Although HEK293T cells had a lower dsODN integration rate, they exhibited a higher off-index than K562 cells, underscoring their proficiency in off-target profiling. iPSCs had the highest off-index for 2 out of 3 sgRNAs, affirming their exceptional sensitivity in off-target event detection (Fig. 5c).

These observations imply that while dsODN integration efficiency is crucial, it is not the only determinant of a cell line's capacity to identify off-target sites. Elements such as chromatin accessibility, which tends to be more open in iPSCs, might influence Cas9's cleavage specificity and subsequently

affect off-target detection. Therefore, dsODN integration at the cleavage site remains pivotal for successfully identifying off-target effects.

**Enhanced off-target detection in iPSCs using constitutive Cas9 expression**

Our study examined the effects of transient versus constitutive Cas9 expression on off-target detection sensitivity. Temporary Cas9 plasmid delivery often leads to delayed enzyme activity and reduced dsODN availability. By contrast, cells with constitutive Cas9 expression experience

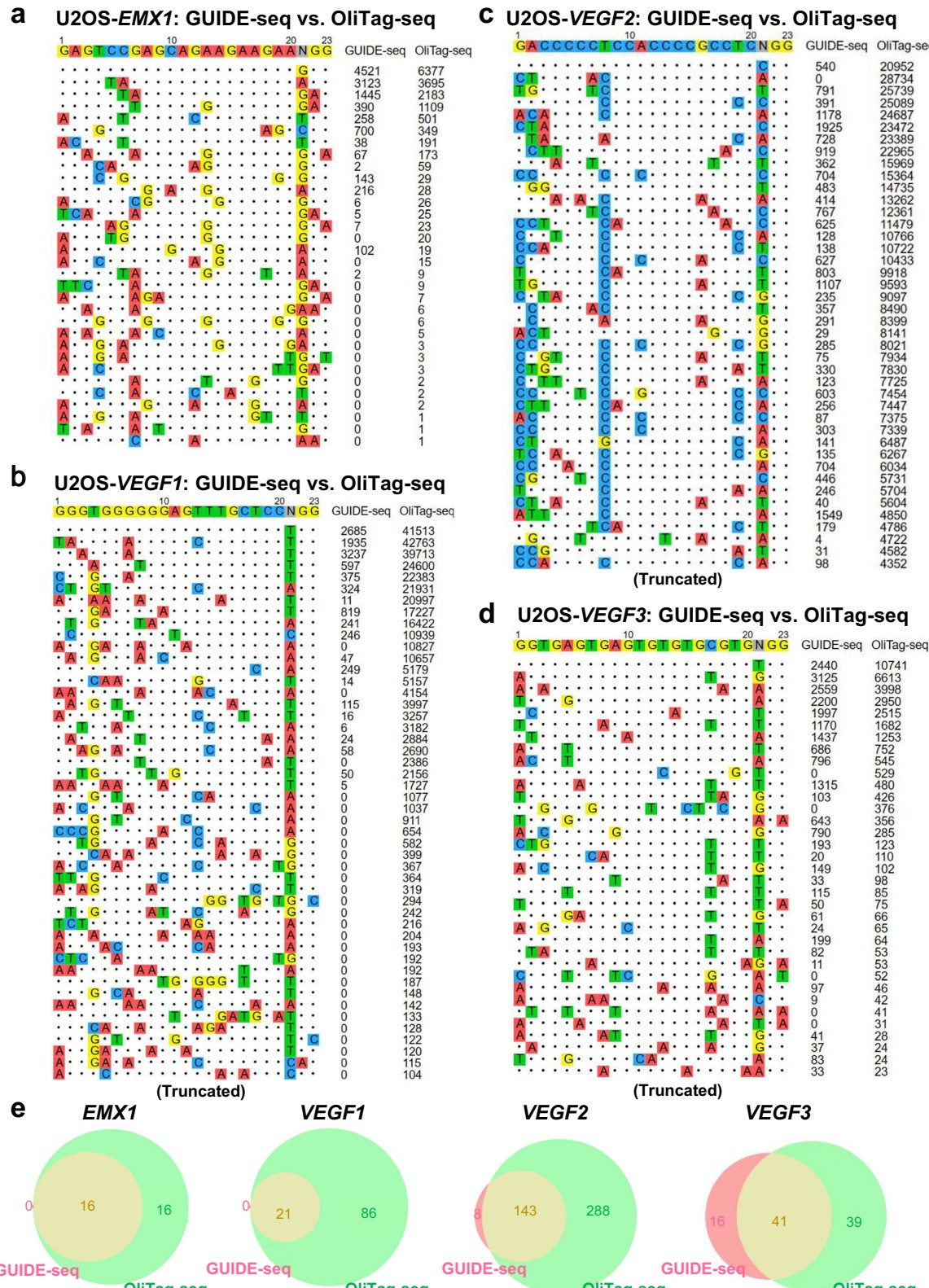

**Fig. 4 | OliTag-seq Sensitivity in Off-Target Detection vs. GUIDE-seq.**
**a–d** Comparative Site Identification: These panels compare off-target sites detected by OliTag-seq and GUIDE-seq for sgRNAs targeting EMX1 (**a**), VEGF1 (**b**), VEGF2 (**c**), and VEGF3 (**d**) in U2OS cells. Next to each listed site, the respective read counts from OliTag-seq are indicated, showcasing the technique's sensitivity. Only a selection of off-targets are shown for VEGF1, VEGF2, and VEGF3, with a comprehensive list available in Supplementary Data 1. The data for GUIDE-seq were obtained from a previous publication. **e** Overlap Analysis of Off-Target Sites: Venn diagrams demonstrate the shared and unique off-target sites identified by GUIDE-seq and OliTag-seq for each sgRNA. This visual comparison underscores the enhanced detection capability of OliTag-seq.

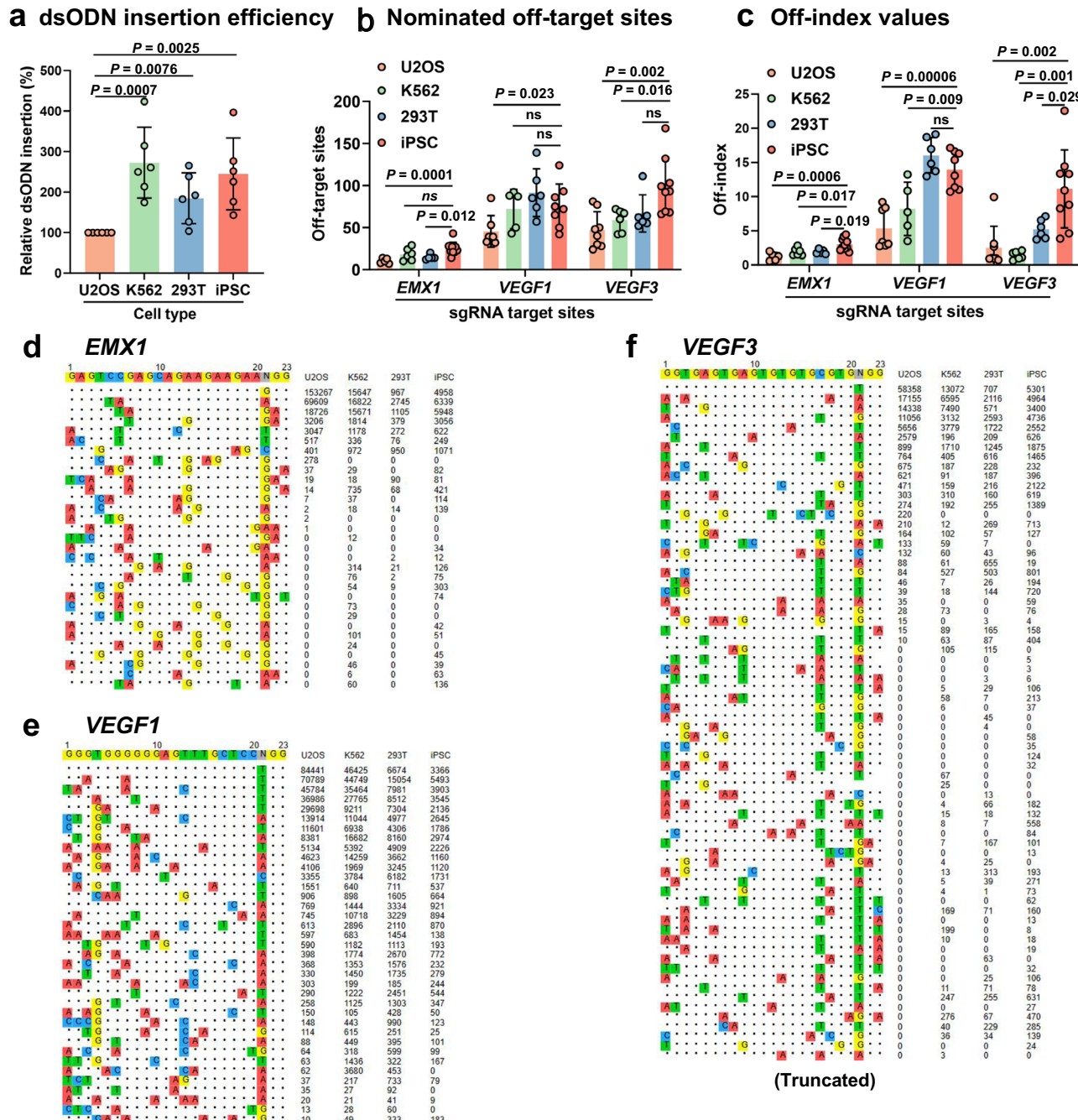

**Fig. 5 | Enhanced Off-Target Detection in iPSCs via OliTag-seq. a** dsODN Insertion Efficiency Across Cell Lines: Presents a comparison of dsODN insertion efficiency among U2OS, K562, HEK293T, and iPSC lines ($n = 6$), with values normalized to U2OS as the reference. **b** Cell Line -Specific Off-Target Site Detection: Enumerates off-target sites identified by OliTag-seq at EMX1, VEGF1, and VEGF3 genomic locations across the four cell types ($n = 5$–10 per site), highlighting variations in detection sensitivity. **c** Off-Index Value Comparison: Evaluates the off-index—a ratio of total off-target to on-target reads—for three distinct sites across U2OS, K562, HEK293T,

and iPSCs ($n = 5$–10 per site), offering insights into the relative precision of off-target detection. **d**–**f** Off-Target Sequences in Cell Lines: Details sequences of off-target sites in the four cell lines for sgRNAs targeting EMX1 (**d**), VEGFA site 1 (**e**), and VEGFA site 3 (**f**). Sequences for VEGF1 and VEGF3 are selectively presented, with complete datasets available in Supplementary Data 2. Data are represented as mean ± s.d., with P values determined by a paired two-tailed Student's $t$ test. Adjusted $P$ values are provided, with "ns" indicating non-significant differences ($P > 0.05$).

immediate sgRNA interaction, continuous DNA cleavage, and improved tagging due to the consistent presence of dsODNs[27,28]. While constitutive expression may increase off-target mutations, especially at less accessible genomic sites, we hypothesize that it significantly bolsters off-target detection.

Our analysis using sgRNAs targeting *EMX1* and *VEGF3* showed efficient on-target cleavage in both transient and constitutive Cas9 expression systems, with sequencing reads predominantly confirming on-target sites

(Fig. 6a, b; Supplementary Data 3). Notably, constitutive Cas9 expression in iPSCs led to a marked increase in off-target cleavage events for both genes, displaying an off/on-target ratio nearly twice that observed with transient expression (Fig. 6c). This trend was similarly observed in K562 cells with stable Cas9 expression.

These results suggest that iPSCs, and possibly other cell types, exhibit significantly improved off-target detection capabilities when employing constitutive Cas9 expression. This enhancement occurs despite the potential

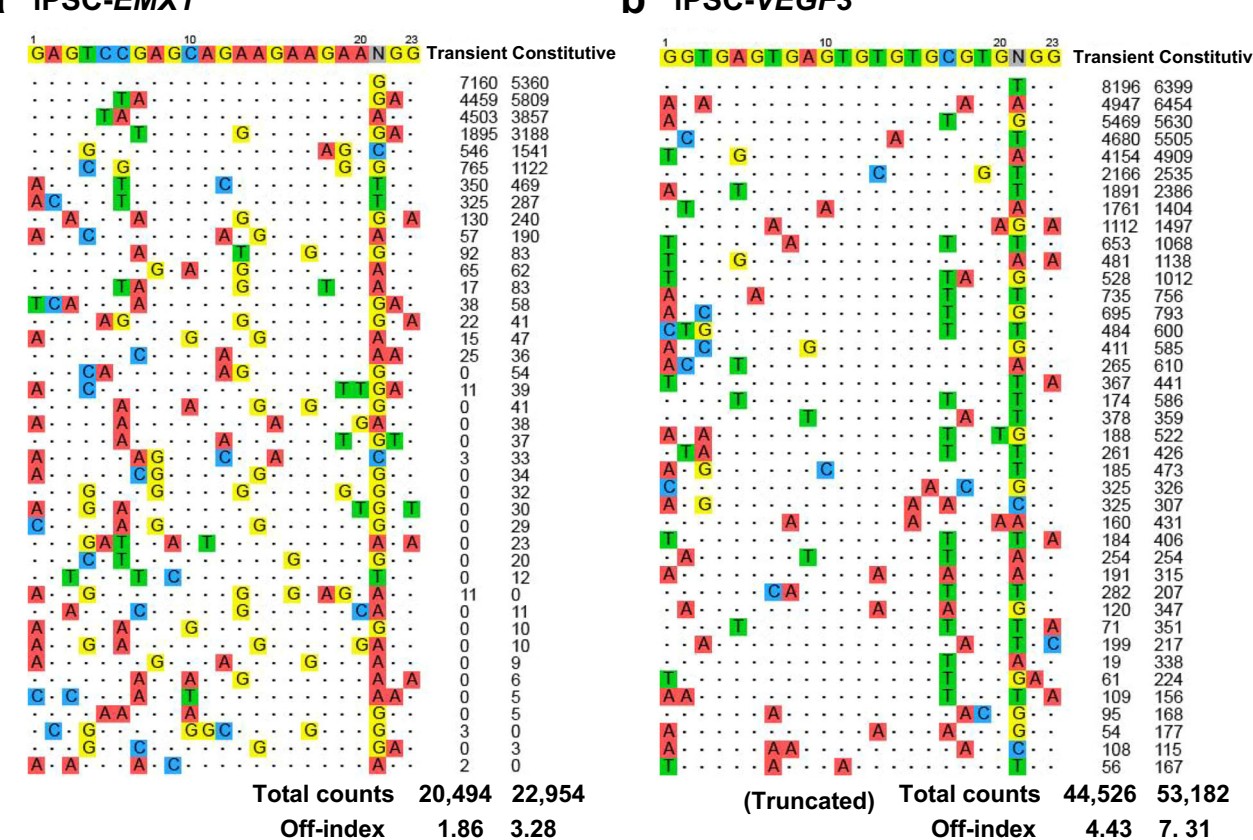

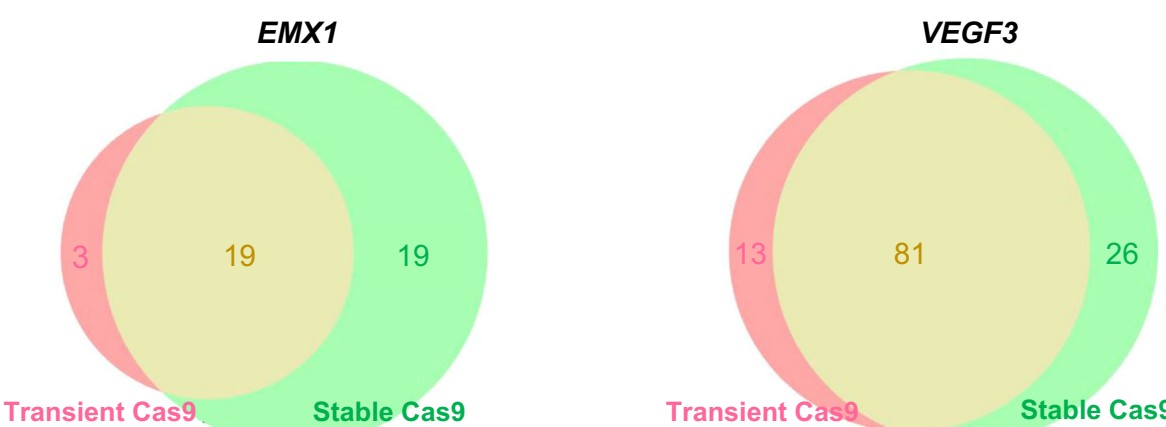

**Fig. 6 | Off-Target Detection Boosted by Stable Cas9 in iPSCs. a** EMX1 Off-Target Site Mapping in iPSCs: Depicts off-target sites for EMX1 in iPSCs, contrasting outcomes from cells with transient versus stable Cas9 expression. **b** VEGF3 Off-Target Visualization in iPSCs: Shows VEGF3 off-target sites in iPSCs, again comparing transient and constitutive Cas9 expression. The sequence details for VEGF3 are abridged, with a comprehensive account in Supplementary Data 3. Off-index values, representing the ratio of off-target to on-target reads, are presented below each visualization. **c** Comparative Off-Target Site Analysis for EMX1 and VEGF3: A Venn diagram summarizes the off-target sites for EMX1 (left) and VEGF3 (right) identified in iPSCs with transient or stable Cas9 expression, illustrating the differential detection profiles.iPSC with constitutive Cas9 expression enhance off-target identification.

uptick in off-target mutations, underscoring the method's effectiveness in comprehensive off-target site identification.

### Enhanced detection of off-target sites in iPSCs through HDAC inhibitor-mediated chromatin modulation

The efficacy of in cellulo off-target assays like OliTag-seq can sometimes be constrained by the limited accessibility of certain genomic regions, an issue that persists even in iPSCs despite their generally more open chromatin compared to differentiated cells. To mitigate this limitation, we explored the use of HDAC inhibitors, specifically Trichostatin A (TSA), which is known to induce chromatin decondensation, potentially enhancing the accessibility of the CRISPR-Cas9 complex to DNA targets[25,29].

In our study, iPSC-Cas9 cells were treated with TSA, and the editing efficiency at various genomic sites was evaluated. These sites included both

transcribed (*AAVS1, CD326, EEF1A1, EEF2*) and untranscribed (*CCR5, CIITA, TRAC, HBG1*) regions, categorized based on transcript levels identified in previous RNA-seq data. Post-TSA treatment, we observed a notable increase in total gene editing efficiency, especially at typically less accessible, closed chromatin sites (Fig. 7a; Supplementary Fig. 9a). The frequency of dsODN integration at these sites rose 1.32-fold in open chromatin and 2.04-fold in closed chromatin (Fig. 7b; Supplementary Fig. 9b). This finding implies that TSA not only boosts the overall efficiency of gene editing but also promotes dsODN integration in regions that were previously less conducive to such modifications. It is important to note that while the state of chromatin may affect where Cas9 generates breaks, the successful identification of these breaks with OliTag-seq is contingent on effective dsODN integration at the cleavage site.

Upon evaluating the sgRNAs targeting *EMX1* and *VEGF3*, it was evident that TSA treatment significantly enhanced the iPSC-Cas9 cell line's capacity for off-target detection. In cells treated with TSA, the number of newly identified off-target sites for *EMX1* and *VEGF3* sgRNAs increased to 16 and 56, respectively (Fig. 7c). To confirm our results' accuracy and dependability and mitigate the risk of false positives, we conducted control experiments using iPSC-Cas9 cells treated with TSA without any sgRNA. Consistently across three biological replicates, these controls yielded zero reads (Supplementary Fig. 9c), reinforcing the specificity and credibility of our findings. This suggests TSA's role in revealing off-target sites that were previously undetectable. However, we noted a decrease in sequencing reads

for certain off-target events following TSA treatment, potentially indicative of TSA-related cytotoxic effects, meriting further exploration.

Comparative analysis of the off-target profiles from TSA-treated iPSC-Cas9 cells with original GUIDE-seq data demonstrated significant enhancement, showing a 2-3 fold increase in the identification of off-target sites for both *EMX1* and *VEGF3* (Fig. 7d). This elevates our TSA-integrated OliTag-seq method as a potentially unparalleled in cellulo assay in terms of sensitivity for off-target detection. It effectively narrows the gap between the high sensitivity of in vitro assays and the inherent limitations of cellular assays. Using TSA and iPSCs in the assay protocol, while potentially altering the off-target landscape, is vital for capturing the most comprehensive range of off-target events. Although these experimental conditions may not fully replicate in vivo scenarios, they provide a thorough assessment of potential off-target risks, vital for further validation in cell types that more accurately represent the physiological context of therapeutic CRISPR-Cas9 applications.

### Precision off-target analysis of sgRNAs in clinical CAR-T cell therapy using OliTag-seq

In clinical settings, especially in CAR-T cell therapies, selecting sgRNAs with minimal off-target impacts is crucial. To evaluate the precision of sgRNAs in such therapeutic contexts, we employed our refined OliTag-seq method on iPSC-Cas9 cells, which served as a permissive iPSC model in cellulo assay on

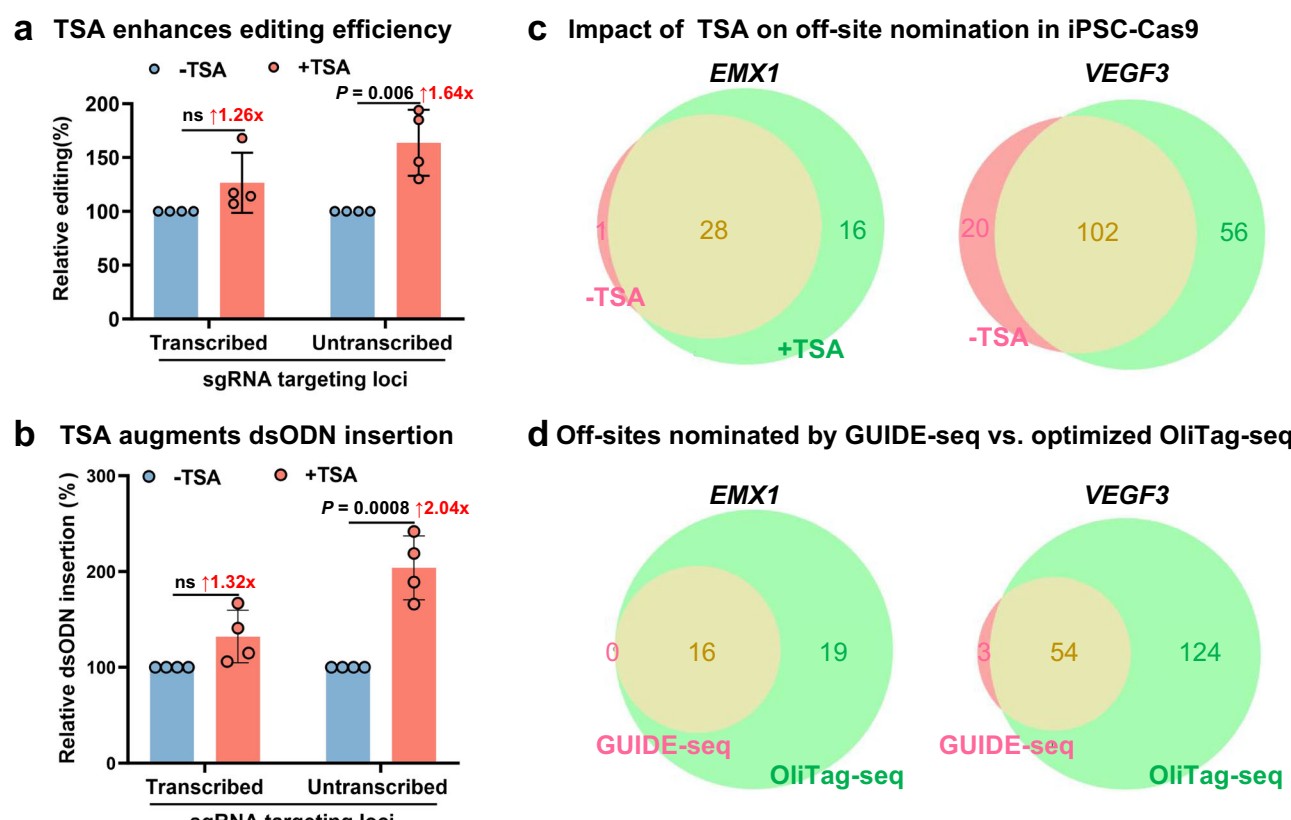

**Fig. 7 | TSA's Role in Boosting Off-Target Detection. a** TSA's Impact on Editing Efficiency: Analyzes TSA's (trichostatin A) effect on editing efficiency at both transcribed (AAVS1, CD326, EEF1A1, EEF2) and untranscribed (CCR5, CllTA, TRAC, HBG1) gene sites (*n* = 4). Transcription activity is quantified using TPM (transcripts per million reads) from RNA-seq data, with TSA significantly improving editing efficiency at less actively transcribed locations. **b** dsODN Insertion Enhancement by TSA: Ilustrates how TSA treatment increases dsODN insertion frequency at the aforementioned gene sites (*n* = 4), with +TSA values normalized to the -TSA control. The increase is notably significant at untranscribed

sites. **c** TSA and Off-Target Detection in iPSC-Cas9: Venn diagrams display the effect of TSA on identifying off-targets for EMX1 (left) and VEGF3 (right) in the iPSC-Cas9 cell line. For VEGF1 and VEGF3, off-target details are included in Supplementary Data 4. **d** Off-Target Site Overlap Analysis: Compares off-target sites identified in iPSC-Cas9 cells with EMX1 (left) and VEGF3 (right) sgRNAs between published GUIDE-seq results and OliTag-seq optimizations. Data are expressed as mean ± s.d., with significance determined by paired two-tailed Student's *t* tests and annotated adjusted P values. The term "ns" is used to denote nonsignificant findings (*P* > 0.05).

therapeutic guide RNA. This study focused on several genes integral to CAR-T therapy, including *TRAC-CJ*, *TRAC-MH*, *TRBC-CJ*, *TRBC-MH*, and *PDCD1-CJ*. We analyzed their off-target effects, adhering to the rigorous standards established by iGUIDE[30], which permit no more than six mismatches in the target or PAM sequences.

Our investigations with OliTag-seq identified 4 to 29 off-target sites per sgRNA, successfully reconfirming five off-target sites previously reported in iGUIDE studies (Fig. 8a–e). Importantly, OliTag-seq also detected additional, previously unrecognized off-target cleavage sites that iGUIDE had not identified. However, it did miss some sites with fewer than ten reads in iGUIDE analysis (Supplementary Fig. 10). Notably, the *TRAC-MH* sgRNA exhibited a higher frequency of off-target mutations. In contrast, the *TRBC-MH* sgRNA demonstrated remarkable specificity, with its off-target reads closely aligning with on-target sequences (Fig. 8b, d). The *PDCD1-CJ* sgRNA showed remarkable specificity, with the number of on-target reads significantly surpassing off-target reads (Fig. 8e). Utilizing hiPSCs as a substitute for primary target cells, despite their different chromatin states compared to differentiated cells such as T-lymphocytes, offers a valuable model for off-target analysis in scenarios where direct testing of primary cells is not feasible.

Moreover, we extended OliTag-seq experiments to primary T cells to identify actual off-targets in CAR-T therapy-related genes (*TRAC-CJ*,

*TRBC-CJ*, *PDCD1-CJ*). Despite some advantages of the longer dsODN used in iGUIDE, it poses challenges, including increased toxicity in primary cells like T cells (Supplementary Fig. 11a). Applying OliTag-seq to primary T cells, we compared side-by-side with iGUIDE. This comparative analysis revealed varying efficacy: OliTag-seq surpassed iGUIDE in detecting off-targets for *TRBC-CJ*, while for *TRAC-CJ* and *PDCD1-CJ*, it identified fewer off-target sites (Supplementary Fig. 11b–e). PCR validation of select off-target sites corroborated the accuracy of OliTag-seq findings (Supplementary Fig. 11f–h).

When contrasting the extensive off-target profiles of the three sgRNAs, collating data from both iGUIDE and OliTag-seq, we observed a comparable total number of off-target sites identified by both methods. While iGUIDE produced a higher total read count, possibly due to more in-depth sequencing, it pinpointed only two high-confidence off-target sites supported by over ten reads. Conversely, OliTag-seq detected more than twenty such sites, indicating superior sensitivity in nominating off-target sites compared to GUIDE-seq and iGUIDE.

In conclusion, our thorough testing of OliTag-seq affirms its high sensitivity and dependability for off-target detection, particularly in the context of CAR-T cell therapy. The method's effectiveness in primary T cells underscores its suitability as a comprehensive tool for risk assessment, enhancing confidence in the safety and efficacy of CAR-T treatments.

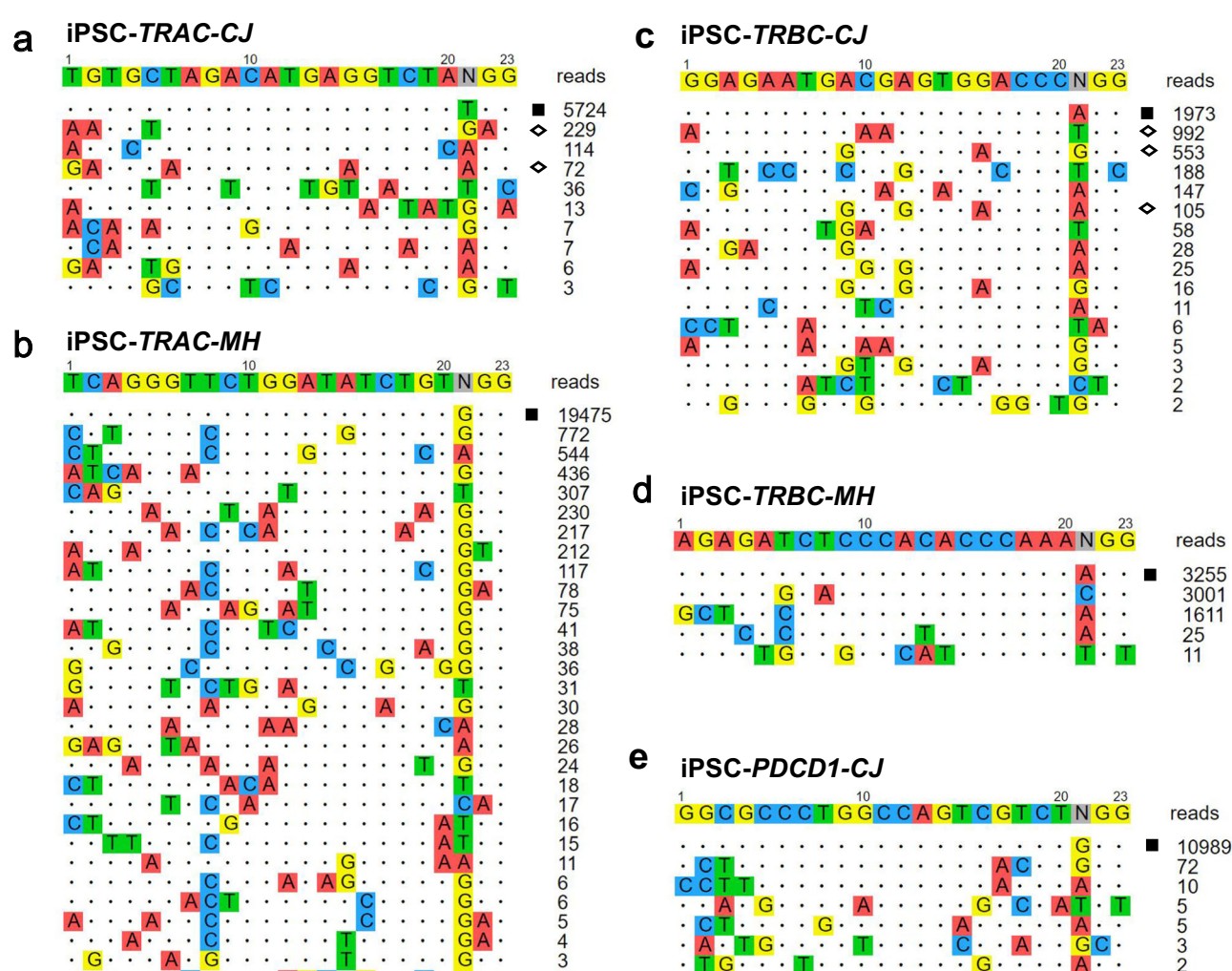

**Fig. 8 | Off-Target Mapping in CAR-T Loci via OliTag-seq. a–e** Depicts off-target detection in CAR-T cell therapy-related genes using OliTag-seq. Shown are sgRNAs aimed at TRAC-CJ (**a**), TRAC-MH (**b**), TRBC-CJ (**c**), TRBC-MH (**d**), and PDCD1-CJ (**e**) in the iPSC-Cas9 cell line using a complex of crRNA:tracrRNA. TSA (0.1 pM) was applied overnight after electroporation to enhance detection. Mismatched sequences at off-target sites are color-coded. Read counts from OliTag-seq are annotated beside each target. On-target sites are denoted with black squares, and off-targets previously identified by iGUIDE are marked with open diamonds.

## Discussion

In our study, we present OliTag-seq, an innovative in cellulo assay that significantly heightens the detection sensitivity of off-target effects induced by CRISPR-Cas9, outperforming the established GUIDE-seq technique. At the core of OliTag-seq's effectiveness is integrating a robust dsODN structure, specifically designed for tagging double-stranded breaks (DSBs) and a unique triple-priming approach in the initial PCR phase. This innovative method facilitates a broader and more comprehensive capture of off-target events compared to traditional PCR techniques. Further enhancing its utility, OliTag-seq incorporates barcodes at the 5' end of the second forward primers. This strategic placement not only streamlines the sequencing process but also offers a cost-effective alternative for high-throughput sequencing projects. A pivotal finding of our research is the exceptional sensitivity of iPSCs in detecting off-target sites, attributable to their globally open chromatin state. Additionally, our results demonstrate that combining constitutive Cas9 expression in iPSCs with HDAC inhibitor treatments substantially improves the detection of off-target modifications. This combination of persistent Cas9 activity, chromatin-modifying agents, and the intrinsic chromatin dynamics of iPSCs underscores the potential of OliTag-seq to enhance the precision and safety of genome editing tools, marking a significant step forward for more accurate and reliable applications in both research and therapeutic contexts.

To optimize dsODN integration at Cas9-induced cleavage sites and enhance junction sequences' amplification, we extended the dsODN length to 39 bp and incorporated GC clamps at both ends. This alteration led to a notable increase in sequencing read counts and more effective detection of off-target events, likely due to improved ligation of the GC-rich dsODN ends with DSBs. During this optimization process, B02 and Mirin emerged as potent enhancers of NHEJ-mediated dsODN insertion, each increasing the efficiency by approximately 1.8-fold, primarily through inhibiting HDR at cleavage sites[31]. The use of B02 is particularly significant, as it may increase the visibility of low-frequency off-target sites. However, its application must be carefully considered, especially for sgRNAs with extensive off-target activity, where B02's cytotoxic effects could become a limiting factor. Consequently, applying B02 in OliTag-seq assays necessitates a nuanced, sgRNA-specific approach, tailored to balance the benefits of enhanced detection against the potential for increased cytotoxicity.

OliTag-seq represents a notable leap forward in cost efficiency for off-target analysis in genome editing. This assay circumvents the need for specialized sequencing protocols that GUIDE-seq requires for platforms like MiniSeq or MiSeq. OliTag-seq is inherently compatible with more economical and high-throughput options, such as the NovaSeq series, primarily due to the relocation of barcodes to the 2nd PCR primers. This adjustment standardizes adaptors across all samples, thus eliminating the expensive requirement of unique adaptors for each sample, a characteristic of the GUIDE-seq approach. OliTag-seq's efficiency is further enhanced by incorporating a triple-priming method in the first PCR stage. This technique allows for the simultaneous amplification of genomic sequences adjacent to the dsODN tag, effectively doubling the yield of junction PCR. This minimizes variability and reduces the resources needed to conduct separate amplification reactions, streamlining the process and cutting costs significantly.

By amplifying both ends of the dsODN insertion in a single reaction, OliTag-seq potentially minimizes biases that might arise if these sequences were amplified separately. This dual-end amplification approach is likely to enhance detection sensitivity by providing a more comprehensive representation of the tagged genomic sequences. The introduction of a triple priming system in the initial amplification phase may lead to a complex mixture of products, but this complexity is efficiently managed in OliTag-seq. The method incorporates a secondary, nested PCR step using just two primers, specifically designed to amplify target regions while diluting or filtering out any non-specific products from the first PCR. By segregating the products of the left and right linkers in this secondary PCR, we ensure clear detection and minimize the potential for complications arising from non-specific products. These strategic optimizations in the PCR process facilitate

the use of commercial sequencing services, significantly reducing overall costs. Consequently, OliTag-seq breaks down economic barriers, making high-sensitivity off-target detection more accessible and affordable.

The enhanced off-target detection capabilities of OliTag-seq are not solely dependent on dsODN integration efficiency; chromatin accessibility also plays a pivotal role. This is particularly relevant in more compact chromatin regions of somatic cells, where off-targets may go undetected[32,33]. iPSCs, with their open chromatin state owing to pluripotency, serve as an optimal model for uncovering potential off-target effects of CRISPR-Cas9 applications. Our comparative analysis between iPSCs and other cell types, including U2OS, K562, and HEK293T cells, demonstrates iPSCs' superior ability to reveal a broader range of off-target events. Consequently, for a more accurate reflection of sgRNA specificity in clinical settings, we advocate using patient-specific iPSCs in off-target screening to accommodate individual genetic variances. This is especially pertinent for primary cells that are sensitive to dsODN or in scenarios of in vivo editing, where conventional tagging and nomination of off-target sites may be challenging.

While plasmid delivery of Cas9 and sgRNA is a cost-effective and widely employed strategy, it may lead to prolonged Cas9 expression, potentially widening the window for off-target alterations. Our study underscores that the constitutive expression of Cas9 can enhance the detection of such off-target events. This aligns with prior observations that increased Cas9 dosage and extended-expression correlate with increased off-target effects[34]. However, transient delivery methods like ribonucleoprotein (RNP) or mRNA can limit Cas9 activity duration, possibly mitigating unintended edits. OliTag-seq's heightened sensitivity is adept at capturing these distinctions, providing a nuanced view that can aid researchers in fine-tuning their gene-editing strategies. Additionally, we found that transiently inducing chromatin opening with HDAC inhibitors facilitates the discovery of off-target sites within typically inaccessible chromatin domains. These techniques increase the accessibility of Cas9 and sgRNA, augmenting the potential for off-target cleavage. We advocate for applying the iPSC-Cas9-TSA model in clinical off-target analysis, as our findings offer key insights into optimizing off-target detection protocols and highlight the importance of considering Cas9 delivery methods in comprehensive off-target evaluations.

In our analysis of CAR-T therapies, OliTag-seq was leveraged to meticulously assess the specificity of Cas9, focusing on genomic sites of clinical importance. Demonstrating enhanced performance over iGUIDE, OliTag-seq not only corroborated established high-confidence off-target sites but also uncovered new sites, enriching our understanding of Cas9's off-target behavior. The methodology surmounted the challenges faced with primary T cells, which often perish when exposed to the 46 bp dsODN used in iGUIDE, complicating off-target detection[35]. While iPSCs are similarly sensitive to Cas9-sgRNA and dsODN complexes, our approach found that their survival is greatly improved with the co-transfection of BCL-XL[36]. Continued research is imperative to validate the off-target sites identified by OliTag-seq in a clinical context and to develop measures that minimize undesirable CRISPR-Cas9 edits. OliTag-seq's capacity for reliable off-target site detection affirms its essential contribution to creating safe and precise CAR-T cell therapies. Notably, employing iPSCs as a model for off-target discovery may unveil sites not observable in T cells due to their distinct chromatin configurations. Nevertheless, initial off-target mapping in iPSCs could pave the way for targeted amplicon sequencing in T cells, thus enhancing the detection process.

We investigated the reproducibility of off-target detection with OliTag-seq and noted that sites with sequencing read counts below 10 showed inconsistent reproducibility. Therefore, we recommend a moderate increase in sequencing depth for critical experiments, particularly those related to clinical research. OliTag-seq demonstrated a higher sensitivity in identifying off-target sites even at lower sequencing depths than GUIDE-seq, indicating its superior detection capability. Moreover, we have validated OliTag-seq's ability to identify off-target sites with mutation frequencies below 0.1% through targeted amplicon sequencing[37]. However, some off-target mutation frequencies may fall below the detection threshold of current high-

throughput sequencing technologies. Increasing the number of initial cells or sample size may help overcome these detection limits.

To circumvent the low sensitivity inherent in GUIDE-seq, hypersensitive in vitro techniques like CIRCLE-seq and CHANGE-seq have been developed. While these methods are adept at initial screenings, they often nominate off-target sites that do not manifest in cellular systems, mainly due to the disparity between purified genomic DNA and the chromatin context within nuclei. Thus, they tend to overestimate potential off-targets, with the majority not being biologically relevant. OliTag-seq, particularly when combined with TSA, narrows the sensitivity gap between cellular assays like GUIDE-seq and these hypersensitive in vitro methods. It serves as a practical approach for confirming the most genuine off-targets from the plethora identified by the initial screenings. Hence, OliTag-seq with TSA not only enhances the detection of true off-target events but also provides a more balanced and realistic representation of the off-target landscape.

Emerging improvements to the original GUIDE-seq, such as Tag-seq, GUIDE-tag, and Extru-seq, have advanced the field of off-target consequence analysis in CRISPR-Cas9 genome editing[38–40]. These methods bring their innovations; however, they also introduce new complexities. For instance, Tag-seq and GUIDE-tag rely on biotinylated ODN integration during DNA repair, which might lead to integration inefficiency and false positives. Extru-seq requires a cell lysis step, adding to experimental intricacy. Moreover, approaches like Tn5 tagmentation or whole genome sequencing, while comprehensive, can incur high costs. OliTag-seq, in contrast, streamlines the off-target discovery process without such complexities and at a lower expense. It enables a nuanced understanding of the off-target profiles associated with various Cas9 delivery methods, guiding the optimization of gene-editing protocols. For gene-edited cell therapies where precision is paramount, OliTag-seq's ability to detect rare off-target events—even with transient Cas9 activity—makes it invaluable for ensuring therapeutic safety. Researchers must weigh these strengths against the limitations of each technique to select the most appropriate method for their experiments.

While this study presents a robust analysis of OliTag-seq's capabilities, it is important to acknowledge its limitations. The focus on a select number of sgRNAs and cell types, although insightful, constrains the breadth of our findings. Future studies could expand the range of sgRNAs and cell types examined, providing a more comprehensive view of CRISPR-Cas9's off-target effects. Moreover, our current OliTag-seq protocol does not account for off-target sites that include base insertions or deletions between the DNA target and the sgRNA, known as bulges or gaps[38]. This calls for enhancing the data analysis algorithm to capture a full spectrum of off-target events from sequencing data.

In conclusion, OliTag-seq has established itself as a method that surpasses GUIDE-seq in detecting a wider array of CRISPR-Cas9 off-target sites. Its ability to integrate with high-throughput sequencing platforms, coupled with its sensitivity in various cell types, including iPSCs, makes it a valuable tool for the gene editing field. The study further underscores the potential of combining constitutive Cas9 expression with HDAC inhibitors to increase the fidelity of off-target detection. These insights are crucial for advancing CRISPR-Cas9 technology development and optimization, ensuring safer and more effective applications. OliTag-seq stands out as a promising avenue for improved off-target analysis, contributing to the overarching goal of precise and responsible use of CRISPR-Cas9.

# Methods and materials
## Gene editing component resources
dsODN preparation. We prepared dsODNs of 34 bp and 39 bp, as depicted in Fig. 1b, using high-fidelity ssODNs sourced from Integrated DNA Technologies (IDT). The annealing protocol commenced with an initial denaturation at 95 °C for 1 s, followed by cooling to 60 °C for another second. This was succeeded by a gradual ramp-down of temperature at approximately −0.05 °C/second to a final hold at 15 °C, ensuring optimal hybridization of the ssODNs into stable dsODNs.

CAR-T-related sgRNA annealing. Custom crRNAs targeting key loci for CAR-T therapy, namely TRAC, TRBC, and PDCD1, along with a universal tracrRNA scaffold, were acquired from IDT. For annealing, both tracrRNA and crRNA were initially reconstituted in 1× TE buffer to a stock concentration of 200 μM. Annealed gRNAs were prepared at a working concentration of 30 μM by mixing crRNA and tracrRNA in a 2:1 ratio, followed by dilution in 5× annealing buffer (Synthego). The annealing process involved heating the mixture to 78 °C for 10 min, then transitioning to 37 °C for 30 min, and finally allowing it to equilibrate to room temperature over 15 min. For electroporation, we utilized 120 pmol of the annealed sgRNA.

Cas9 and sgNRA expressing plasmids construction. We designed sgRNAs targeting EMX1, VEGF1, VEGF2, and VEGF3 based on sequences reported in the GUIDE-seq study[15]. Additional sgRNAs aimed at human AAVS1, CD326, EEF1A1, EEF2, CCR5, CIITA, TRAC, and HBG were developed using the CHOPCHOP platform[41]. All sgRNAs commenced with a guanine nucleotide to activate the U6 promoter, which is essential for transcriptional efficiency. The sequences used in this study are detailed in Table S1.

For plasmid construction, we employed the NEBuilder HiFi DNA Assembly Master Mix (New England Biolabs), following previously established protocols[42,43]. Each fragment was PCR-amplified from existing plasmids in our collection using KAPA HiFi polymerase (KAPA Biosystems) and purified with the Zymoclean Gel DNA Recovery Kit (ZYMO Research). The fragments were then seamlessly assembled into a plasmid backbone using the NEBuilder® HiFi DNA Assembly Cloning Kit. Clones were screened by Sanger sequencing (Tsingke Biotechnology) and Nanopore sequencing (GenoStarBio) to confirm the correct assembly.

## Human cell culture and electroporation
U2OS cells. U2OS cells, obtained from Procell Life Science & Technology, were maintained in Iscove's Modified Dulbecco's Medium (IMDM; Gibco) supplemented with 10% fetal bovine serum (FBS; Gibco) at 37 °C in a 5% $CO_2$ atmosphere. The media was refreshed bi-weekly. For electroporation, cells were suspended in Solution SE (Lonza) and subjected to the Lonza 4D-Nucleofector program DN-100 or in Amaxa™ Cell Line Nucleofector™ Kit V solution (Lonza) using the Lonza 2b Nucleofector program X-001, per manufacturer's guidelines.

In the process of validating off-target effects, $1 \times 10^6$ U2OS cells were electroporated with the gene-editing components (1 μg of Cas9 expression plasmid and 1 μg of sgRNA-encoding plasmid) but without incorporating dsODN. Genomic DNA (gDNA) was extracted from the U2OS cells three days post-electroporation. PCR primers were designed to target around 28 off-target sites identified for EMX1 and VEGF3 in the OliTag-seq analysis. DNA fragments of approximately 250 bp, encompassing the sequences of these off-target sites, were amplified using PCR. The combined PCR products were then prepared for 150 bp paired-end sequencing.

K562 cells. K562 cells, sourced from the ATCC, were cultured in Advanced RPMI 1640 medium (Gibco) with 10% FBS, maintained at 37 °C with 5% $CO_2$, and passaged routinely according to cell density. Electroporation was conducted using the Amaxa™ Cell Line Nucleofector™ Kit V following program T-016.

HEK293T cells. HEK293T cells, also from ATCC, were cultured in DMEM (Gibco) supplemented with 10% FBS and incubated at 37 °C with 5% $CO_2$ in a humidified environment. These cells were electroporated with the Amaxa™ Cell Line Nucleofector™ Kit V using the A-023 program.

Human iPSCs. Human iPSCs were derived from peripheral blood mononuclear cells of an anonymous donor using established reprogramming factors[44,45]. Culture plates were pre-coated with Matrigel (Corning) before iPSC seeding. The cells were maintained in mTeSR™ E8 medium (Gibco), with daily medium changes. For electroporation, iPSCs

were processed using the Human Stem Cell Nucleofector™ Kit 2 (Lonza) and program B-016. Routine passaging involved gentle disaggregation into small clusters with a 0.5 mM EDTA/PBS solution (Thermo Fisher Scientific) and distributed into new wells without reducing to single cells.

In the off-target analysis, iPSCs underwent electroporation with both the editing components and dsODN. Approximately 1-1.5×10⁶ cells were dissociated into a single-cell suspension using Accutase (Innovative Cell Technologies). The cell pellet was then resuspended in 70 μl of electroporation solution containing 1 μg of Cas9 expression plasmids, 1 μg of sgRNA encoding plasmids, and 100 pmol of dsODN. Additionally, 0.5 μg of BCL-XL transient expression plasmids were included for iPSCs to enhance cell viability post-electroporation, complemented with 10 μM of the ROCK inhibitor Y-27632 (LC Labs)[36]. When necessary, iPSCs were also treated with 0.1 μM TSA (Trichostatin A) overnight after electroporation.

To assess the false positive background of OliTag-seq, a similar protocol was followed: (1–1.5) × 10⁶ iPSCs were electroporated with the 39 bp dsODN, but without any sgRNA. The cell pellet was resuspended in 70 μl of the same electroporation solution, supplemented with 0.5 μg of BCL-XL transient expression plasmids and 10 μM of Y-27632 and 0.1 μM TSA (Trichostatin A) applied overnight post-electroporation. Three days following electroporation, gDNA from the iPSCs was extracted, and the OliTag-seq library was prepared.

**Human primary T cells.** We cultured 1×10⁶ human primary T cells in ImmunoCult™-XF T Cell Expansion Medium (StemCell), supplemented with 10% FBS, 20 ng/ml IL-2, 5 ng/ml IL-7, 5 ng/ml IL-15, and 20 μl CTS™ Detachable Dynabeads™ CD3/CD28 beads (Thermofisher). The CD3/CD28 beads were employed for T cell activation. Following a three-day activation period, the T cells were resuspended in 20 μl of P3 electroporation solution (Lonza) containing 1 μg of SpCas9 protein, 120 pmol of sgRNA, and 50 pmol of dsODN. The cells were then electroporated using the Lonza 4D-Nucleofector with program EH-115, following the manufacturer's instructions. Three days post-electroporation, genomic DNA (gDNA) was extracted from the T cells, and the OliTag-seq library was prepared for off-target analysis.

For off-target verification, 2×10⁶ T cells were electroporated with the gene-editing components (1 μg of SpCas9 protein and 120 pmol of sgRNA), but without dsODN. After three days, gDNA was extracted from these cells. PCR primers targeting approximately eight off-target sites identified for *TRAC-CJ*, *TRBC-CJ*, and *PDCD1-CJ* from the OliTag-seq results were designed. DNA fragments of about 250 bp, encompassing the sequences of these off-target sites, were amplified via PCR. The combined PCR products were subsequently prepared for 150 bp paired-end sequencing.

To assess the toxicity of dsODN of varying lengths, T cells were electroporated solely with three types of dsODN (34 bp, 39 bp, 46 bp), respectively. A group without dsODN served as a control. On the third day post-electroporation, the cell count for each group was recorded, providing insight into the potential toxicity related to different dsODN lengths.

### Establishment of iPSC-Cas9 cell line

For generating iPSC-Cas9 cell lines, iPSCs were plated at a density of 1×10^5 cells per well in a Matrigel-coated 24-well plate with 500 μl of medium. A lentiviral vector co-expressing SpCas9 and a puromycin resistance gene was added to the culture at a multiplicity of infection (MOI) of 1, alongside 0.1% Poloxamer synperonic F108 (Sigma) and 5 μg/ml Protamine sulfate (Sigma-Aldrich) to enhance viral transduction[46,47]. After 24 hours, the medium was refreshed. Cas9-puro expressing cells were selected with 1 μg/ml puromycin treatment overnight, three days post-transduction.

For electroporation, 1-1.5×10^6 iPSC-Cas9 cells were transfected with either 1 μg of sgRNA plasmid or 120 pmol of annealed sgRNA together with 100 pmol of dsODN, negating the need for a Cas9 expression plasmid.

### Genomic DNA isolation and OliTag-se q library preparation

Three days post-electroporation with the gene editing and tagging reagents, genomic DNA (gDNA) was extracted using the Gentra Puregene Blood Kit (Qiagen) following the manufacturer's protocol. The quantity of gDNA was determined using Qubit fluorometry (Invitrogen).

For library preparation, 1 μg of gDNA was diluted to 130 μl with 1× TE buffer (IDT) and sheared to an average length of 500-700 bp using a Covaris S220 instrument. The sheared DNA was then purified with 1.5 volumes of Select-a-Size DNA Clean & Concentrator MagBeads (Zymo Research). Following this, end-repair, A-tailing, and adaptor ligation steps were performed using the KAPA Hyper Prep Kit (KAPA Biosystems), according to the kit's instructions. The adaptor includes an 8 nt random molecular index to facilitate unique identification of each DNA fragment. After ligation, DNA fragments were purified again to remove unligated adaptors and dimers using a 1.2× bead cleanup.

The purified, adaptor-ligated DNA underwent two rounds of nested PCR amplification using KAPA HiFi DNA Polymerase, specifically targeting the sequences flanking the dsODN-tagged sites. The resultant PCR products were pooled and prepared for 150 bp paired-end sequencing on an Illumina platform, as conducted by Novogene.

A comprehensive protocol for OliTag-seq, including detailed reagent concentrations and PCR conditions, is available in the Supplementary Protocol section.

### Identification of Cas9-induced off-target cleavage sites using OliTag-seq pipeline

We used a customized all-in-one OliTag-seq pipeline to analyze the 150 bp paired-end Illumina sequencing data output. The pipeline would align the filtered reads to the reference human genome and retain potential off-target sequences, allowing for seven or fewer mismatches in both target and PAM sequences. To install and conduct the OliTag-seq pipeline, please follow the subsequent steps.

  a. Installation of OliTag-seq Software Pipeline:
    It is recommended to set up a dedicated Python environment for OliTag-seq using Anaconda. The necessary environment configuration file (environment.yaml) is available for download from our GitHub repository (https://github.com/qwe1234567891/OliTag-seq/archive/refs/heads/main.zip). Use the following commands to install and activate the OliTag-seq environment:
    *unzip main.zip*
    *cd OliTag-seq-main*
    *conda env create -f environment.yml*
    *conda activate olitag*
  b. Setup of Human Genome Reference:
    Download the reference human genome fasta file from the provided GitHub link.
  c. Creation of a Configuration File:
    A.yaml configuration file should be prepared containing the raw data storage path, sample barcode sequences, and paths for the required tools. The OliTag-seq software package on GitHub includes detailed examples and instructions.
  d. Run the Analysis:
    With the conda environment activated, initiate the analysis using the command:
    *python OliTag-seq/OliTag.py all -m manifest.yaml*

### Targeted deep sequencing to validate editing frequencies

To verify indel frequencies and dsODN insertion rates at the targeted sites, we employed barcoded PCR, as previously detailed[25]. Using 200 ng of genomic DNA per reaction, we amplified regions spanning 200-280 bp around the target sites. The KAPA HiFi DNA polymerase was utilized for amplification, and the resulting samples were sequenced with a 150 bp paired-end Illumina platform. Data demultiplexing and analysis were conducted using Seqkit and Barcode-splitter[48]. The editing frequencies and dsODN insertion rates were quantitatively assessed with CRISPResso2[25,49]. Relevant PCR primer sequences are provided in Supplementary Data 5.

## dsODN insertion rate analysis

A proprietary algorithm was developed to quantify dsODN insertion frequencies post-demultiplexing of sequencing data. Reads bearing target sequences were earmarked for further evaluation. Those lacking dsODN insertions were categorized as 'unedited', while those featuring insertions were extracted via the grep command for 'edited' categorization. We applied specific criteria to differentiate between single and double dsODN insertions: for 34 bp dsODNs, reads exceeding the reference sequence length by 29-49 bp indicated a single insertion, and an increase of 64–84 bp suggested a double insertion. Similarly, for 39 bp dsODNs, a length excess of 34-54 bp denoted a single insertion, whereas an extension of 69–89 bp was indicative of double insertions. The insertion frequency was deduced by summing the counts of reads exhibiting either one or two dsODN insertions.

For assessing full-length dsODN insertions, reads were analyzed for the presence of either the 34 or 39 nt oligos in their entirety, in both forward and reverse orientations. Full-length dsODN insertion rates were calculated by dividing the tally of reads with complete insertions by the aggregate of reads with any insertions, including those with truncated oligos.

The Python script for this analytical algorithm is an integral part of the OliTag-seq software suite, ensuring a streamlined process for users.

## Statistics analysis

Data analysis was performed using GraphPad Prism 8.0.1 software. Normal distribution was assumed for all data, and statistical significance was evaluated with paired or unpaired two-tailed Student's $t$ tests. Results are presented as mean ± SEM, and $P$ values less than 0.05 were considered statistically significant.

## Statistics and reproducibility

The experimental design and statistical methodologies applied across various data analyses in this study are detailed within respective sections of the results and methods. Specifically, for OliTag-seq, a minimum of three independent biological replicates were utilized, encompassing different target sites across various cell lines. Statistical significance was assessed using paired two-tailed Student's $t$ tests, with significance levels annotated via adjusted p-values.

## Reporting summary

Further information on research design is available in the Nature Portfolio Reporting Summary linked to this article.

## Data availability

The 150PE Illumina sequencing raw data are deposited in the Sequence Read Archive (SRA) database under accession number PRJNA967129 and are publicly accessible at https://www.ncbi.nlm.nih.gov/sra/PRJNA967129. The source data behind the graphs in Figs. 7 and 3 can be found in Supplementary Data 4 and 6 respectively.

## Code availability

The OliTag-seq software pipeline for off-target sites nomination and the Python code for dsODN insertion rate analysis are publicly available on GitHub (https://github.com/qwe1234567891/OliTag-seq).

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

## Acknowledgements

This work was supported by the National Key Research and Development Program of China (Grant Nos. 2019YFA0110803, 2019YFA0110204, and 2021YFA1100900), the National Natural Science Foundation of China (Grant Nos. 82070115 and 81890990), the Chinese Academy of Medical Sciences (CAMS) Innovation Fund for Medical Sciences (CIFMS) (Grant Nos. 2022-I2M-2-003, 2022-I2M-2-001, 2021-I2M-1-041, 2021-I2M-1-040, and 2021-I2M-1-001), the Nonprofit Central Research Institute Fund of Chinese Academy of Medical Sciences (Grant No. 2020-PT310-011), the Tianjin Synthetic Biotechnology Innovation Capacity Improvement Project (Grant No. TSBICIP-KJGG-017), the CAMS Fundamental Research Funds for Central Research Institutes (Grant No. 3332021093), the Haihe Laboratory of Cell Ecosystem Innovation Fund (Grant No. HH23KYZX0005 and HH22KYZX0022), the State Key Laboratory of Experimental Hematology Research Grant (Grant No. Z23-05), and the Postdoctoral Fellowship Program of CPSF (Grant No. GZB20230081).

## Author contributions

X.B.Z., Z.X.Y., and Y.W.F. designed the experiments and contributed to the concepts. Z.K.Z. contributed to bioinformatic pipeline development and high-throughput sequencing data analysis. Z.X.Y. and D.H.D. were involved in data analysis and interpretation. Z.X.Y. and F.Z. cloned the plasmids. W.W., and X.L. conducted the CAR-T-related experiments. Z.X.Y. and X.B.Z. wrote the manuscript. D.H.D., Z.Y.G., and X.B.Z. revised the manuscript. J.P.Z. and H.Y.L. provided administrative support and supervision. All authors approved the article for submission and publication.

## Competing interests

The authors declare no competing interests.
