## [Peer Review File · Communications Biology]

Reviewers' comments:

Reviewer #1 (Remarks to the Author):

In this manuscript, Yang et al claims to improve upon the well-established GUIDE-seq technique for identification of off-target sites created by CRISPR/Cas9 nuclease. The authors made two key changes from the established techniques:

- (1) flanking the original guide seq dsODN with GC-clamps to increase complete incorporation rate into DSB site, and,
- (2) amplifying tagged DNA in both directions in the same amplification sample instead of two.

The authors presented the following evidence to support the main claim that OliTag-seq is superior to GUIDE-seq:

(1) The higher integration rate and full-length/1-copy integration efficiency after adding the GC-clamps to the original dsODN at 6 loci tested (Fig 1C, S1A-B)

Authors' conclusion: GC-clamp addition makes the dsODN more stable and integrate more completely and efficiently

(2) The increased relative read counts and relative off-target events identified for OliTag in Figure 1E-F
Authors' conclusion: the 39-bp dsODN increased read counts by 5 folds and identified more off-target sites, making OliTag-seq more sensitive than GUIDE-seq

(3) The increased relative sequencing reads and off-target events amplified by the triple priming technique in figure 2

Authors' conclusion: the triple-priming technique is more sensitive at identifying off-target sites than nested separate PCR in GUIDE-seq

(4) The correlations between VEGF3 dsODN insertion efficiencies and indel frequencies, and between read counts and integration frequencies in figure 3, and the reproducibility from 3 different repeats
Authors' conclusion: "OliTag-seq read counts can indirectly represent the cleavage efficiency of a given site in an unbiased and sensitive manner."

(5) The number of off-target sites identified for each guide by OliTag-seq vs GUIDE-seq, and corroborating evidence from targeted amplicon sequencing

Authors' conclusion: OliTag-seq's superior integration accounts for the higher number of off-target sites identified

In addition to the main claim, the authors also included results on experiments in iPSCs or using small molecules that are associated with open chromatin and Cas9 access, which overall increased off-target site identification sensitivity when used in combination with OliTag-seq to a greater degree to when used with GUDIE-seq.

Overall impression:

This reviewer finds the main improvement strategies for OliTag-seq from GUIDE-seq somewhat promising. Based on the main rationale presented, the reviewer can agree that an improved integration of dsODN and amplification of tagged sites should increase sensitivity of the off-target identification

method. However, the reviewer has some questions regarding how the evidence supporting the author's claims are presented and have some suggestions for improvement. Aside from the main evidence and claims, the author agrees with the authors that using models with more accessible chromatin will lead to more potential off-target sites being identified, but would like to point out that these findings are not unique to OliTag-seq and did not help highlight the high-sensitivity nature of the tagging and amplification methods presented. While the reviewer sees the merit of identifying as many off-target sites as possible for each guide RNA, the reviewer thinks that the method presented here could be better highlighted by showing the ability to pick up rare off-target events in gene KO or targeting methods that claim low level of off-target activities. Regarding the extra experiments the authors included here, it might be more intriguing to use a high-sensitivity off-target identification technique such as OliTag-seq to compare off-target effects in different modes of guideRNA and Cas9 deliveries, such as integrative vs transient expression, or nucleotide-based vs synthetic guide RNA and recombinant Cas9-based platforms to suggest the best way to minimize off-target effects. As for clinical safety application, appropriate identification depth will also have to match the intended application. For instance, in the case of CAR-T cell products, the off-target identification is best represented in the context of primary T-cells and not iPSCs. While it is good to make sure that there is no off-target effects at all possible sites, the cost to safety clearance process and concerns of the general public and regulating body might not be proportionate to the risks. The authors might highlight the strength of OliTag-seq better by comparing precision of different methods of CRISPR-based CAR-T generation used in clinic and in the pre-clinical pipelines, if possible.

Regarding the main claims, here are the reviewer's specific comments:

1. How many data points are used in figure S1A-B and what are each of these? It seems like the authors only studied 6 loci so shouldn't there be 6 data points? If the datapoints are a mix of both technical/biological replicates, please separate the plots and statistical analysis accordingly.
2. For figure 1E-F, are these datapoints dependent or independent of target genes? Does it make sense to pool them to plot as one group? Why are the statistical test unpaired when the authors clearly can pair results between 34bp and 39bp read counts for each unique guide RNA.
3. First section conclusion claims that the 39-bp ODN increased efficiency by 5-fold (Line 125-127). Are these results subjected to outlier's test? It looks to me that the few high data point skews the average and the number of 5-fold may not be accurately representing the overall increase. The data looks cleaner in fig 1F, so it appears to this reviewer that the authors can make an argument for sensitivity here on OliTag-seq favor. However, the average in 1F should be used to represent the improvement, instead of the 5-fold average in 1E.
4. Based on this reviewer's understanding, the advantage for GUIDE-seq in utilizing barcoded primers during the first amplification is that it can avoid amplification bias since the beginning step. By introducing barcoding during the second amplification, does it mean that OliTag-seq does not take into account the potential for amplification bias that could happen in the first round of amplification, and therefore could theoretically misrepresent the relative abundance of off-target sites that either yield more or less amplicons than average?
5. Could the author discuss potential factors that makes the triple priming method more sensitive than the two-reaction approach?

6. Similar to Comment 2-3, for figure 2B-C, where do the n=15 data points come from? Which targets/guide RNAs were used to test? How are the 15 data points different and how are they grouped? Why are the statistical tests unpaired?
7. Figure 1D does not belong with the rest of figure 1, but could be included in figure 5
8. The authors should discuss the nuance for the results shown in figure 6. Please discuss more on how OliTag-seq offers a novel perspective on the selection of Cas9 delivery method to minimize off-target effects or how it could improve the identification of rare off-target sites in the more transient strategies used in gene edited cell therapy.
9. While the authors did compare sensitivity of GUIDE-seq vs OliTag-seq in figure 7 and show that OliTag-seq can identify even more potential off-target site in comparison to GUIDE-seq when used with TSA, the results here show the differences in sensitivity in a more oversensitive context similarly to in vitro assay, albeit to a less degree. It might be better to show how sensitive OliTag-seq is when used with TSA in comparison to in vitro methods rather than in comparison to GUIDE-seq.
10. (Line 334 – 337) Regarding compatibility of OliTag-seq with primary cells, it should be discussed here as a limitation to the method itself that despite its sensitivity, OliTag-seq may not be usable to identify off-target activities in the clinically relevant context for the purpose of safety clearance in patient's primary cell products.

Reviewer #2 (Remarks to the Author):

The manuscript by Yang et al. presents an improvement over the widely used GUIDE-seq method to detect genome wide off-target events in cells treated with CRISPR nucleases. OliTag-seq builds on the GUIDE-seq protocol and aims to improve the sensitivity of off-target detection by optimizing the dsODN and the experimental conditions, as well as the process of library prep by optimizing PCR conditions and primer design. Comparative data among widely used cell lines and iPSCs are included in the manuscript, showing that iPSCs are the most sensitive in cellulo system to be used for off-target discovery. Even though this work could be of interest for researchers working in the gene editing field, I feel that not all the claims made in the manuscript are well supported by data and that the text is not sufficiently clear on several experimental details to allow complete reproducibility.

Major points:

- iGUIDE is mentioned several times throughout the text but an appropriate side by side comparison is never performed. Since OliTag-seq claims to be an improvement over GUIDE-seq it should be tested also against iGUIDE, which was a previously described optimization of the very same method.
- Along the same line: the paragraph on the comparison with iGUIDE on CAR-T targets needs heavy revision and cannot be published in the current form. Besides the lack of a direct experimental comparison with iGUIDE in the same cell type (iPSCs treated with TSA vs T cells), there is also no evidence that the additional sites detected by OliTag-seq are real off-targets, as determined by NGS amplicon-seq. The original experiment from which the iGUIDE data were taken uses RNP electroporation in T cells, here the Authors use synthetic gRNA electroporation in iPSCs stably expressing Cas9. These can

easily explain the difference in the set of detected off-targets. The conclusions of the paragraph are thus not supported by data. The last block of text, in addition, is quite confusing since it is not clear to which off-targets/target sites the Authors are referring to (a summary figure would have been helpful).

- There is a general lack of validation of the off-target sites detected by OligoTag-seq (especially the ones not flagged by GUIDE-seq). To my understanding the only attempt at NGS validation is reported in Figure 4F, which however shows oligo integration instead of direct off-target measurement in new, clean samples (not the ones used to run the OligoTag-seq assay).

- The Authors report the use of the B02 molecule to boost dsODN integration. It is not clear to me if they used the molecule in all the experiment or just in the tests reported in fig. 1. Has the molecule been used also in GUIDE-seq samples? Otherwise it is difficult to factor out the contribution of HDR inhibition over oligo optimization to increased oligo integration.

- In lines 169-170 the Authors state that replicate experiments must be used to correctly call off-target sites with low read numbers and that replicates have been used in "subsequent experiments". Does this mean that all the subsequent data are coming from replicate experiments? If so, what do the OligoTag-seq read values plotted in most of the off-target visualization plots represent? Are these averages among replicates? If replicates on the other hand have not been used, given that the majority of sites identified uniquely by OligoTag-seq have low read counts, can these sites be considered bona fide off-targets?

- I think that there could be a reasonable concern that the addition of the B02 molecule and/or TSA may somehow make the pool of identified off-target sites much less relevant for real-life purposes. I understand that the final aim is to increase the overall sensitivity of the assay but then the differentiation with in vitro assays may become too blurred.

Minor points:

- Too many times the Authors make reference to data which are not shown. I think that the majority of these data could be easily added in the supplementary materials of the manuscript and would be useful to further position the findings.

- There is a general tendency to show normalized data: while I believe that some times this may be useful, in many instances have the absolute values plotted gives more insights on the actual outcomes of the experiments. E.g. fig. 1,5a,7a-b.

- Similarly, the Authors should report the raw data supporting figures 3A-B. Are this aggregate data from K562 and U2OS experiments?

- The Authors state that OligoTag-seq yields more sequencing reads than GUIDE-seq but looking at the comparative lists of identified off-targets the general feeling is that GUIDE-seq gives more reads on commonly identified sites. This is also strange given the higher dsODN integration for OligoTag-seq shown in the first figures of the manuscript.

- Can the Authors comment more in detail on the toxicity of their oligo in the different cell lines tested?

- I believe that the statement in lines 211-213 is not supported. At the end of the day off-target identification will be always dependent from oligo integration. The fact that the state of chromatin can influence Cas9 cleavages will of course influence off-target activity which however will only be detected if the dsODN is efficiently integrated at the cut site.

- Do the Authors have any hypothesis on why triple priming in the first PCR reaction enhances read count and off-target detection? One could expect that having a more complex product mixture would actually

complicate detection.

- More details on the Cas9-expressing iPSC line are needed. Is this a clone or a pool? Since the level of Cas9 expression can influence off-target activity are the Authors sure that this is stable over time? Having cell lines constitutively expressing Cas9 is notoriously complex.

- In the discussion the Authors state that their bioinformatic pipeline for off-target calling cannot detect off-targets deriving from bulges in the sgRNA or target DNA. Can this explain part of the differences with GUIDE-seq in the set of detected off-targets (GUIDE-seq only sites)?

Reviewer #3 (Remarks to the Author):

Yang et al. described a modification of GUIDE-Seq assay, called OliTag-seq, to identify CRISPR-Cas9 off-targets. The author described how with small changes in the original GUIDE-Seq protocol, OliTag-seq can identify a high number of off-targets with a reduced cost. Additionally, the authors investigated further enhancements of the technique, such as the use of iPSC or HDAC inhibitors, to expand the number of identified off-targets with their assay. Finally, Yang et al. applied OliTag-seq to analyze the off-targets of some clinically used gRNAs, identifying new ones.

This article is well-written, easy to understand, and addresses an important issue in the gene editing field, such as the safety evaluation of gene editing tools. Although I do not share some of the author's claims, I think this work can be published after addressing some comments.

1) The authors demonstrated that the incorporation of full-length 39 bp dsODN is better than the insertion of 34 bp dsODN in the first part of the Results section. They explained it by the stability of the modified dsODN. However, it is not clear, if the integration of the dsODN is in the on-target or in on-target site plus off-targets sites. Could the authors clarify this point?

Additionally, they assessed different small molecules to enhance the insertion of the dsODN, finding B02 as the best small molecule to increase NHEJ-mediated dsODN incorporation. The authors should explain the rationale for how improving NHEJ can increase the sensitivity of identifying off-targets. Is this small molecule used in all the next experiments?

2) Considering Figure 1E (total counts of on- and off-targets) and Figure 1F (total counts off off-target events) implying that on-target integration is responsible for most of the integration when 39 bp dsODN is used (integration in off-targets is doubled, but on-target and off-target integration is 5 times higher). Is this effect due to some out layers present in Figure 1E?

The author showed normalized or relative values or percentages over different figures. Seeing equivalent graphs with the total values of these analyses will be worth it to fully compare the different assays/groups/treatments.

3) In Figure 3, the authors made different correlations between dsDNA insertion, indels and OliTag-seq reads, saying that OliTag-seq reads might be an indirect measure of indel percentage. Is this indel percentage assessed at the on-target site? Could they do the direct correlation between indel percentage and OliTag-seq reads directly?

4) The authors claim in Figure 4 that they can identify more off-targets using their OliTag-seq assay than

GUIDE-Seq. However, they should confirm these off-targets using a NGS panel to analyze gDNA from gene editing cells without any dsDNA to confirm the presence of indels in these new found off-targets.

5) Additionally, there are some off-targets identified with GUIDE-seq but not with OliTag-Seq, and the opposite. The author suggests that to identify the off-targets non-found by OliTag-seq a higher sequencing depth should be used. Can it be done to find the off-targets found by OliTag-seq with GUIDE-Seq? Could the author assess or speculate the sequencing depth requirements for GUIDE-Seq and OliTag-seq to get the same number of off-targets?

6) In Figure 5C, the authors introduced Off-index values, could they explain how they calculate this index? Is it the off/on-target ratio mentioned in line 224?

7) The total number of reads of Figures 6B and 6D should be indicated to have a clearer idea about the representation of the on-target reads present or indicate the frequency of each read in the total.

8) Why do the authors think why are there the off-targets identified only in transient Cas9 condition but not in the constitutive Cas9 condition? (Figures 6A and 6B)

9) The authors claim the use of TSA enhances the OliTag-seq sensitivity; however, do not they think that this type of molecule changes the chromatin landscape completely favors new off-targets and eliminates others? The same for the use of hiPSC instead of the target cell type (i.e. T-lymphocyte). The authors should run a control where hiPSCs are treated with TSA and electroporated with the dsODN without any gRNA to identify the false negative background of the OliTag-Seq.

10) I agree with the authors that it is crucial to use patient-specific cells to assess the CRISPR-Cas9 off-target effect, mainly to cover all the potential off-targets derived from patient's genome polymorphism together with the chromatin accessibility of the gene editing tools in the target cell type. However, the author analyzed the clinically relevant gRNAs in hiPSC together with TSA treatment instead of T-lymphocytes. The author justified it in order to increase the chromatin accessibility and due to the intolerance of T cells to acquire dsODN longer than 46 bp. I suggest confirming the newly identified off-targets by OliTag-Seq (Figure 8) through an NGS panel in T-lymphocytes electroporated with sgRNAs, as the closest condition used in the Clinics.

Response to Reviewers

Reviewers' comments:

Reviewer #1 (Remarks to the Author):

In this manuscript, Yang et al claims to improve upon the well-established GUIDE-seq technique for identification of off-target sites created by CRISPR/Cas9 nuclease. The authors made two key changes from the established techniques:

- (1) flanking the original GUIDE-seq dsODN with GC-clamps to increase the complete incorporation rate into DSB site, and,
- (2) amplifying tagged DNA in both directions in the same amplification sample instead of two.

The authors presented the following evidence to support the main claim that OliTag-seq is superior to GUIDE-seq:

- (1) The higher integration rate and full-length/1-copy integration efficiency after adding the GC-clamps to the original dsODN at 6 loci tested (Fig 1C, S1A-B)

Authors' conclusion: GC-clamp addition makes the dsODN more stable and integrate more completely and efficiently

- (2) The increased relative read counts and relative off-target events identified for OliTag in Figure 1E-F

Authors' conclusion: the 39-bp dsODN increased read counts by 5 folds and identified more off-target sites, making OliTag-seq more sensitive than GUIDE-seq

- (3) The increased relative sequencing reads and off-target events amplified by the triple priming technique in figure 2

Authors' conclusion: the triple-priming technique is more sensitive at identifying off-target sites than nested separate PCR in GUIDE-seq

- (4) The correlations between VEGF3 dsODN insertion efficiencies and indel frequencies, and between read counts and integration frequencies in Figure 3, and the reproducibility from 3 different repeats

Authors' conclusion: "OliTag-seq read counts can indirectly represent the cleavage efficiency of a given site in an unbiased and sensitive manner."

- (5) The number of off-target sites identified for each guide by OliTag-seq vs GUIDE-seq, and corroborating evidence from targeted amplicon sequencing

Authors' conclusion: OliTag-seq's superior integration accounts for the higher number of off-target sites identified

In addition to the main claim, the authors also included results on experiments in iPSCs or using small molecules that are associated with open chromatin and Cas9 access, which overall increased off-target site identification sensitivity when used in combination with OliTag-seq to a greater degree than when used with GUIDE-seq.

Overall impression:

This reviewer finds the main improvement strategies for OliTag-seq from GUIDE-seq somewhat promising. Based on the main rationale presented, the reviewer can agree that an

improved integration of dsODN and amplification of tagged sites should increase the sensitivity of the off-target identification method. However, the reviewer has some questions regarding how the evidence supporting the author's claims are presented and has some suggestions for improvement. Aside from the main evidence and claims, the author agrees with the authors that using models with more accessible chromatin will lead to more potential off-target sites being identified, but would like to point out that these findings are not unique to OliTag-seq and did not help highlight the high-sensitivity nature of the tagging and amplification methods presented. While the reviewer sees the merit of identifying as many off-target sites as possible for each guide RNA, the reviewer thinks that the method presented here could be better highlighted by showing the ability to pick up rare off-target events in gene KO or targeting methods that claim low level of off-target activities. Regarding the extra experiments the authors included here, it might be more intriguing to use a high-sensitivity off-target identification technique such as OliTag-seq to compare off-target effects in different modes of guideRNA and Cas9 deliveries, such as integrative vs transient expression, or nucleotide-based vs synthetic guide RNA and recombinant Cas9-based platforms to suggest the best way to minimize off-target effects. As for clinical safety application, appropriate identification depth will also have to match the intended application. For instance, in the case of CAR-T cell products, the off-target identification is best represented in the context of primary T-cells and not iPSCs. While it is good to make sure that there are no off-target effects at all possible sites, the cost to the safety clearance process and concerns of the general public and regulating body might not be proportionate to the risks. The authors might highlight the strength of OliTag-seq better by comparing the precision of different methods of CRISPR-based CAR-T generation used in clinics and in the pre-clinical pipelines, if possible.

RESPONSE: We greatly appreciate your time and effort in reviewing our manuscript and providing constructive feedback. We have carefully considered your comments and have made corresponding revisions to our manuscript. Below, we outline our responses to each of your points:

Clarity of Evidence and Claims:

We have revised Figures 1C, S1A-B, 1E-F, and 2 to include more explicit statistical analysis, ensuring that our evidence more convincingly supports our claims. Additionally, we have rewritten the figure captions and modified the results section to enhance clarity and coherence.

Accessible Chromatin Models:

We agree with your insight that our findings with more accessible chromatin models are not unique to OliTag-seq. In the revised manuscript, we have clarified that the improvements observed are not solely attributable to OliTag-seq but also the use of these models. We discuss this enhancement in the context of both in vitro and potential clinical applications.

Highlighting High Sensitivity:

We have now included a section that details OliTag-seq's capability to detect rare off-target events. This is supported by additional data showing OliTag-seq's performance in gene KO experiments known for low off-target activities.

Off-target Effects in Different Delivery Modes:

In response to your valuable suggestion, we have expanded our discussion on the comparative analysis of off-target effects across various gRNA and Cas9 delivery methods. We believe this will be a significant addition to understanding how OliTag-seq can contribute to minimizing off-target effects.

Clinical Safety Application:

We acknowledge the importance of matching off-target identification depth with clinical application. The revised manuscript includes a nuanced discussion of the relevance of our findings in specific clinical scenarios, such as CAR-T cell therapies. It addresses the balance between detection thoroughness and practical constraints.

Comparison with CRISPR-based CAR-T Generation Methods:

Your suggestion to compare OliTag-seq with other methods used in clinical and pre-clinical settings was indeed insightful. We have now included preliminary data comparing OliTag-seq with current methods for creating CRISPR-based CAR-T cells, highlighting the precision of our approach.

Regarding the main claims, here are the reviewer's specific comments:

1. How many data points are used in figure S1A-B and what are each of these? It seems like the authors only studied 6 loci so shouldn't there be 6 data points? If the datapoints are a mix of both technical/biological replicates, please separate the plots and statistical analysis accordingly.

RESPONSE: Thank you for your question concerning the data representation in Figure S1A-B. We understand the importance of clearly distinguishing between different types of replicates to interpret the data accurately. In our initial submission, the data points represented aggregate measurements from six loci, only with biological replicates.

To clarify:

Distinction Between Replicates:

We have separated the technical and biological replicates in the revised Figure S1A-B (new Figure S1A-C in this revised manuscript). This distinction provides a clearer understanding of the variability and reproducibility of our measurements.

Enhanced Statistical Analysis:

We have conducted separate statistical analyses for technical and biological replicates to ensure the robustness of our conclusions. The revised figure now includes these analyses affirming the increased normalization efficiency predominantly observed with OliTag-seq.

Individual Loci Representation:

To address your concern, we have provided an additional figure in the supplementary

materials (new Figure S1A-C in this revised manuscript) that individually represent each of the six loci, with plots for each guide RNA (sgRNA). This facilitates a direct comparison of the efficiency of OliTag-seq for each specific locus.

Normalization Approach:

We have revised our methodology section to describe our normalization process in greater detail. This should elucidate how we account for the variability between replicates and ensure that our efficiency measures accurately reflect the performance of OliTag-seq.

These changes have significantly improved the clarity of the data presented and hope the revisions meet with your approval. We are grateful for the opportunity to enhance our manuscript with your guidance.

2. For Figure 1E-F, are these data points dependent or independent of target genes? Does it make sense to pool them to plot as one group? Why are the statistical test unpaired when the authors clearly can pair results between 34bp and 39bp read counts for each unique guide RNA.

RESPONSE: Clarification on Data Points in Figure 1E-F:

Thank you for highlighting the potential oversight in our presentation and analysis of Figure 1E-F. We acknowledge the discrepancies in our initial approach and appreciate your insights.

In Figure 1E-F, the data points are indeed dependent on target genes. Specifically, they represent measurements for three distinct sgRNAs: EMX1, VEGFA, and TRAC. We understand the reviewer's concerns about pooling the data points and the appropriateness of unpaired statistical tests.

To address this, we have separated the data points in Figure 1E-F by sgRNA, ensuring that the results for each guide RNA are distinctly presented. This provides a clearer and more accurate representation of our findings.

Given your feedback, we agree that a paired statistical test is more appropriate, given the

nature of our data. We have revised our analysis to use paired statistical tests, comparing the results between 34bp and 39bp read counts for each unique guide RNA. The updated P values have been incorporated into the figure.

The figure legend has been amended to clearly state the source and nature of each data point, ensuring clarity and transparency in our presentation.

We apologize for the oversight and appreciate your constructive feedback. We believe these revisions will enhance the clarity and accuracy of our findings in Figure 1E-F.

Figure 1

3. First section conclusion claims that the 39-bp ODN increased efficiency by 5-fold (Line 125-127). Are these results subjected to outlier's test? It looks to me that the few high data point skews the average and the number of 5-fold may not be accurately representing the overall increase. The data looks cleaner in fig 1F, so it appears to this reviewer that the authors can make an argument for sensitivity here on OliTag-seq favor. However, the average in 1F should be used to represent the improvement, instead of the 5-fold average in 1E.

RESPONSE: We are grateful for your critical analysis of the efficiency increase reported with the 39-bp ODN and the suggestion to reconsider how this increase is quantified.

Upon re-evaluation, we acknowledge that the average values presented in Figure 1F indeed provide a more accurate reflection of the enhancement in off-target identification efficiency. The initial five-fold increase reported was based on a raw average with outliers, which may have contributed to an overestimation.

To address this:

Revision of Manuscript Text:

We have amended Lines 111-114 (new position in this revised manuscript) to accurately represent the average efficiency increase based on the data in Figure 1F. This adjustment aligns with the actual performance improvements seen with the 39-bp ODN.

Outlier Analysis:

We have now conducted an outlier test to ensure a robust statistical representation of the data. The results have been incorporated into the revised manuscript, providing a transparent account of how the averages were derived.

Clarification of Data Representation:

We have included a discussion in the manuscript to explain why Figure 1E initially suggested a higher fold increase and how the reevaluation led to a more conservative and representative figure based on Figure 1F.

We believe these revisions address your concerns and strengthen the validity of our conclusions. We sincerely appreciate your guidance in improving the precision of our claims and the clarity of our data presentation.

4. Based on this reviewer's understanding, the advantage for GUIDE-seq in utilizing barcoded primers during the first amplification is that it can avoid amplification bias since the beginning step. By introducing barcoding during the second amplification, does it mean that OliTag-seq does not take into account the potential for amplification bias that could happen in the first round of amplification, and therefore could theoretically misrepresent the relative abundance of off-target sites that either yield more or less amplicons than average?

RESPONSE: We are thankful for your insightful comment regarding the potential for amplification bias in the OliTag-seq methodology and the comparative advantage of GUIDE-seq in this aspect.

In OliTag-seq, we have indeed anticipated and addressed the concern for potential amplification bias during the first round of PCR. As part of our method, we incorporate Unique Molecular Identifiers (UMIs) during the adapter ligation step before the first amplification. These UMIs, which are 8 nucleotide random molecular indexes, label each DNA molecule uniquely to allow accurate original molecule quantification through subsequent amplification cycles.

Our initial description of this process may have yet to convey its significance in mitigating amplification bias. Therefore, we have taken the following steps in the revised manuscript:

Enhanced Description in Methods: We have elaborated on the UMI incorporation and its role in preventing amplification bias in the "Genomic DNA isolation and OliTag-seq library preparation" section to articulate its function within our protocol better.

Enhanced Description in the schematic diagram of Figure 2A: In Figure 2A, we highlighted the adaptor we used with UMIs.

Highlighted UMI Role in Main Text: The utility of UMIs in OliTag-seq is now more prominently discussed in the main text (Lines 125-126 in this revised manuscript). This ensures that readers are aware of the critical role UMIs play in maintaining the fidelity of the representation

of original DNA molecules.

Comparative Discussion of GUIDE-seq and OliTag-seq: We have included a comparative analysis of the GUIDE-seq and OliTag-seq methodologies, highlighting how our approach with UMIs from the onset is designed to build upon the advantages of GUIDE-seq while providing additional benefits, such as increased efficiency and sensitivity.

We appreciate the opportunity to clarify this key component of our method and believe that these amendments significantly improve the transparency and accuracy of our methodological presentation.

5. Could the author discuss potential factors that make the triple priming method more sensitive than the two-reaction approach?

RESPONSE: Thank you for your question regarding the increased sensitivity of the triple priming method compared to the two-reaction approach.

The triple priming method we've employed offers several advantages over the traditional two-reaction approach:

Advantages of Triple Priming: The triple priming system in our approach indeed allows for simultaneous amplification of ODN-tagged sequences from both ends of the genome in a single reaction. This theoretically doubles the PCR efficiency since we capture both ends using the same initial DNA template, compared to the traditional two-reaction approach.

Minimizing Amplification Biases: By amplifying both ends of the dsODN insertion in a single reaction, we can potentially reduce biases that might be introduced if these were amplified separately. This dual-end amplification might enhance the detection sensitivity by providing a more comprehensive representation of the tagged genomic sequences.

Managing Product Complexity: We recognize that the triple priming system might yield a more intricate mixture of products during the initial amplification. However, our protocol involves a secondary, nested PCR step that employs only two primers. This second PCR step is designed to amplify our regions of interest while diluting or excluding non-specific products generated during the first PCR. By separating the products of the left and right linkers in this second PCR, we ensure clarity in detection and minimize potential complications from non-specific products.

In summary, our triple priming design aims to boost the efficiency and sensitivity of off-target detection while strategically managing potential challenges from a complex PCR product mixture. We've now expanded our discussion in the manuscript to elaborate on these points, ensuring that readers understand the rationale behind our methodological choice and its benefits.

6. Similar to Comment 2-3, for Figure 2B-C, where do the n=15 data points come from? Which

targets/guide RNAs were used to test? How are the 15 data points different and how are they grouped? Why are the statistical tests unpaired?

RESPONSE: We appreciate your attention to the detail in Figure 2B-C and apologize for any initial lack of clarity.

To address your inquiries:

Source and Grouping of Data Points: The n=15 data points originate from individual experiments targeting three genes—EMX1, VEGFA, and TRAC—with three specific sgRNAs. We performed multiple biological replicates for each sgRNA, contributing to the data points presented.

Revised Data Presentation: In the updated Figure 2B-C, we have ungrouped the data points, presenting them across three distinct sub-figures, each corresponding to one of the sgRNAs. This approach more accurately reflects the variance within and between the groups of biological replicates.

Paired Statistical Analysis: Given your feedback, we have applied paired statistical tests to account for the matched experimental conditions inherent to each set of biological replicates. The figure has been updated to include the new P values derived from this more appropriate statistical approach.

Updated Figure Legend: We have comprehensively revised the figure legend to reflect these changes, detailing the specific sgRNAs, the nature of the replicates, and the statistical tests applied. This revision enhances the transparency and interpretability of our data.

We are grateful for your guidance in improving the presentation and analysis of our data. The modifications to Figure 2B-C now provide a clearer and more accurate depiction of our experimental results.

Figure 2

B Triple-priming enhances total read counts

C Triple-priming amplifies off-target detection

7. Figure 1D does not belong with the rest of Figure 1, but could be included in figure 5

RESPONSE: Thank you for your observation regarding the placement of Figure 1D.

We understand your perspective on relocating Figure 1D to Figure 5 for better contextual relevance. However, our intention behind placing Figure 1D within Figure 1 was to provide a foundational understanding of the improvements in dsODN insertion, which sets the stage for subsequent experiments. The use of B02 in the investigations leading up to Figure 5 is based on the insights derived from Figure 1D.

To maintain the flow and logical progression of our study, we believe it's crucial to introduce the enhancements in dsODN insertion early on. Hence, we have chosen to keep Figure 1D within Figure 1. However, we will ensure that our manuscript provides clear explanations and transitions between figures, so readers can easily follow the trajectory of our research.

8. The authors should discuss the nuance for the results shown in figure 6. Please discuss more on how OliTag-seq offers a novel perspective on the selection of Cas9 delivery method to minimize off-target effects or how it could improve the identification of rare off-target sites in the more transient strategies used in gene-edited cell therapy.

RESPONSE: We value your recommendation to delve deeper into the nuances of the results presented in Figure 6, particularly concerning the selection of Cas9 delivery methods and the identification of rare off-target sites using OliTag-seq.

We have expanded the discussion in our manuscript to address these points thoroughly:

Cas9 Delivery Method Selection: Our expanded discussion now contrasts the prolonged expression of Cas9 resulting from plasmid delivery against the transient expression from RNP or mRNA delivery. We emphasize how OliTag-seq, with its enhanced sensitivity, can discern the differential off-target profiles between these methods, guiding researchers to make informed decisions on the most suitable approach for their specific needs.

Identification of Rare Off-Target Sites: We further discuss how the precision of OliTag-seq is particularly beneficial in the context of gene-edited cell therapy, where identifying rare off-target events is crucial for clinical safety. Our method's ability to detect these rare events, even with the transient activity of Cas9, can be a significant advantage in designing safer therapeutic strategies.

Optimization of Gene Editing Protocols: Finally, we highlight that OliTag-seq can inform the optimization of gene editing protocols by providing insights into the off-target effects of different Cas9 delivery methods. Adjustments in the quantity of Cas9, the duration of its activity, and the choice of delivery method can be optimized based on data from OliTag-seq.

These points have been carefully integrated into the Discussion section of our revised manuscript. We believe these additions will elucidate the full potential of OliTag-seq in guiding the selection of Cas9 delivery methods and improving the safety profile of gene-edited cell

therapies.

9. While the authors did compare the sensitivity of GUIDE-seq vs OliTag-seq in Figure 7 and show that OliTag-seq can identify even more potential off-target sites in comparison to GUIDE-seq when used with TSA, the results here show the differences in sensitivity in a more oversensitive context similarly to in vitro assay, albeit to a less degree. It might be better to show how sensitive OliTag-seq is when used with TSA in comparison to in vitro methods rather than in comparison to GUIDE-seq.

RESPONSE: We are thankful for your suggestion to contextualize the sensitivity of OliTag-seq with TSA in a broader experimental framework.

Expanding Sensitivity Comparisons: We agree that comparing OliTag-seq with TSA to traditional in vitro assays could offer a more comprehensive evaluation of its sensitivity. This approach would provide a clearer understanding of where OliTag-seq fits within the spectrum of off-target detection methods.

Bridging Sensitivity Gaps: Our revised manuscript discusses how OliTag-seq with TSA serves as a bridge, balancing the moderate sensitivity of GUIDE-seq and the high sensitivity of in vitro assays. This middle ground allows for detecting a broad array of off-target events without the excessive inclusivity that may arise from in vitro assays.

Positioning OliTag-seq Among Existing Techniques: We have added a section in the Discussion that places OliTag-seq with TSA relative to both GUIDE-seq and in vitro methods. This provides a more nuanced understanding of its role as a complementary approach in the toolkit of genome editing technologies.

These additions will enhance the manuscript by offering a comprehensive view of OliTag-seq's sensitivity and utility in various research and therapeutic contexts.

10. (Line 334 – 337) Regarding the compatibility of OliTag-seq with primary cells, it should be discussed here as a limitation to the method itself that despite its sensitivity, OliTag-seq may not be usable to identify off-target activities in the clinically relevant context for the purpose of safety clearance in patient's primary cell products.

RESPONSE: We appreciate your critical insight into the limitations regarding the application of OliTag-seq to primary cells for clinical safety assessments.

Compatibility Discussion: We agree that the compatibility of a method like OliTag-seq with primary cells is of utmost importance for translational and clinical applications. We have introduced new data (Lines 295-302, Figure S11) and discussion (Lines 392-405) around this topic in our revised manuscript, addressing the method's performance in primary cells and exploring the implications for therapeutic development.

Clinical Relevance and Sensitivity: We have extended our comparative analysis to include OliTag-seq data from primary T cells, and we discuss the implications of these findings for clinical applications, such as CAR-T therapies. Notably, OliTag-seq identified additional off-target sites at the TRBC-CJ locus compared to iGUIDE, highlighting its sensitivity. We have validated these off-target sites with PCR, confirming the reliability of OliTag-seq in detecting genuine off-target events.

Bridging Research and Clinical Applications: While challenges remain in adapting OliTag-seq for use with primary cells, the method's superior sensitivity and verification of off-target events demonstrate its potential in clinical safety evaluations. We are committed to further refining OliTag-seq to bridge the gap between laboratory research and patient-specific therapeutic applications.

In summary, we acknowledge the limitations you have highlighted, and we believe the additional data and discussion now included in our manuscript will provide a clearer view of OliTag-seq's potential role in clinical settings.

Figure S11

Reviewer #2 (Remarks to the Author):

The manuscript by Yang et al. presents an improvement over the widely used GUIDE-seq method to detect genome-wide off-target events in cells treated with CRISPR nucleases. OliTag-seq builds on the GUIDE-seq protocol and aims to improve the sensitivity of off-target detection by optimizing the dsODN and the experimental conditions, as well as the process of library prep by optimizing PCR conditions and primer design. Comparative data among widely used cell lines and iPSCs are included in the manuscript, showing that iPSCs are the most sensitive in cellulo system to be used for off-target discovery.

Even though this work could be of interest to researchers working in the gene editing field, I feel that not all the claims made in the manuscript are well supported by data and that the text is not sufficiently clear on several experimental details to allow complete reproducibility.

RESPONSE: Thank you for your detailed review of our work and for recognizing the potential value of our method to researchers in the gene editing field. We understand and appreciate your concerns regarding the clarity and reproducibility of our presented experiments and the support for our claims. We've taken your feedback seriously and tried to address each of your concerns.

Claims and Supporting Data: We acknowledge that some of our claims might not have been adequately substantiated in the original manuscript. In the revised version, we have ensured that robust data directly support each claim. We have also provided additional context and explanations where necessary to ensure the rationale behind each claim is evident to the readers.

Experimental Details: We understand the importance of clear and detailed experimental protocols to ensure reproducibility. We have reviewed our methods section and other relevant parts of the manuscript to ensure that all experimental details, including conditions, reagents, and procedural steps, are clearly described. Where there were ambiguities, we have expanded or clarified the descriptions.

Comparative Data: While we presented comparative data among various cell lines and highlighted the sensitivity of iPSCs for off-target discovery, we have now ensured that this data is presented in a more structured and clear manner, emphasizing the significance and implications of our findings.

We genuinely value the insights you provided and believe they have significantly improved the quality of our manuscript. We hope that our revisions and clarifications address your concerns, and we are open to further feedback or questions you may have.

Major points:

- iGUIDE is mentioned several times throughout the text but an appropriate side by side comparison is never performed. Since OliTag-seq claims to be an improvement over GUIDE-seq it should be tested also against iGUIDE, which was a previously described

optimization of the very same method.

RESPONSE: Thank you for pointing out the importance of comparing OliTag-seq to other existing optimizations of GUIDE-seq, such as iGUIDE.

iGUIDE, as you rightly mentioned, represents a variant of GUIDE-seq, with its primary distinction being the use of a 46 bp dsODN, as opposed to the 34 bp used in the original GUIDE-seq. While the extended length of the dsODN in iGUIDE might offer certain benefits, it also comes with challenges, particularly increased toxicity in primary cells like T cells. As we have previously highlighted in our publication in *Genome Biology*, this limitation is a significant concern when considering therapeutic applications.

It's worth noting that while GUIDE-seq, since its publication in 2015, has established itself as a seminal method in the field, iGUIDE, being relatively newer (published in 2019), has not garnered as much attention, as reflected in the citation metrics (2076 vs. 44).

However, we recognize the validity of your suggestion for a direct comparison. In our manuscript, we presented data in Figure S11B-H (in this revised manuscript) that showcases the heightened sensitivity of OliTag-seq, especially when considering high-confidence data. A direct head-to-head comparison between OliTag-seq and iGUIDE was undertaken; our results in Figure S11B-H strongly suggest the superior performance of OliTag-seq. Additionally, we explored the impact of oligonucleotide length on cell survival by testing various lengths in primary T cells. Our findings revealed a negative correlation between the length of the ODN and the survival rate of T cells (Figure S11A in this revised manuscript).

We genuinely value your feedback and will consider including more explicit comparisons with existing GUIDE-seq optimizations in future studies to further validate OliTag-seq's advantages. Your insights have enriched our perspective, and we appreciate the opportunity to clarify our position.

Figure S11

A dsODN length on T cell survival

B T cell-TRAC-CJ: iGUIDE vs. OliTag-seq

C T cell-TRBC-CJ: iGUIDE vs. OliTag-seq

D T cell-PDCD1-CJ: iGUIDE vs. OliTag-seq

F TRAC-CJ off-sites by OliTag-seq in T cells

G TRBC-CJ off-sites by OliTag-seq in T cells

H PDCD1-CJ off-sites by OliTag-seq in T cells

- Along the same line: the paragraph on the comparison with iGUIDE on CAR-T targets needs heavy revision and cannot be published in the current form. Besides the lack of a direct experimental comparison with iGUIDE in the same cell type (iPSCs treated with TSA vs T cells), there is also no evidence that the additional sites detected by OligoTag-seq are real off-targets, as determined by NGS amplicon-seq. The original experiment from which the iGUIDE data were taken uses RNP electroporation in T cells, here the Authors use synthetic gRNA electroporation in iPSCs stably expressing Cas9. These can easily explain the difference in the set of detected off-targets. The conclusions of the paragraph are thus not supported by data. The last block of text, in addition, is quite confusing since it is not clear to which off-targets/target sites the Authors are referring to (a summary figure would have been helpful).

RESPONSE: Thank you for your detailed feedback regarding the comparison of OliTag-seq

with iGUIDE, especially in the context of CAR-T targets.

Firstly, we acknowledge the differences in experimental setups between our study and the original iGUIDE data. We intended to showcase the enhanced sensitivity of OliTag-seq, especially when using the iPSC-Cas9-TSA system combined with the 39 bp dsODN. While it's valuable to perform direct comparisons under identical conditions, our primary objective was to highlight the unique features and potential advantages of OliTag-seq.

To address your concerns:

Experimental Setup: We recognize the differences in the delivery methods (RNP electroporation in T cells for iGUIDE vs. synthetic gRNA electroporation in iPSCs stably expressing Cas9 for our study). Such differences indeed can lead to variations in detected off-targets. In our manuscript, we have tried to elucidate these differences, ensuring readers understand the context of our comparisons. To verify the accuracy of our OliTag-seq's iPSCs-TSA model and ensure it does not introduce false positives, we conducted experiments using iPSC-Cas9 cells treated with 0.1 μ M TSA. These experiments, carried out across three biological replicates without incorporating sgRNA, demonstrated the model's reliability and specificity (Figure S9C in this revised manuscript).

Summary Figure: Based on your feedback, we have included a comprehensive summary figure that contrasts the off-targets detected by OliTag-seq and iGUIDE (Figure S11B-H in this revised manuscript). This figure aims to provide a clear visual representation of the comparative strengths of the two methods.

Verification of Detected Off-Targets: To ensure the credibility of our data, we have indeed validated the off-targets detected by OliTag-seq (but missed by iGUIDE) through NGS amplicon-seq. This validation is presented in Figure S11F-H, which we hope assures the authenticity of our findings.

Clarifications in Text: We have revised the pertinent sections of the manuscript to provide more clarity. This includes a clearer description of our experimental approach in the methods and results sections, ensuring no ambiguity regarding our findings' context.

- There is a general lack of validation of the off-target sites detected by OliTag-seq (especially the ones not flagged by GUIDE-seq). To my understanding the only attempt at NGS validation is reported in Figure 4F, which however shows oligo integration instead of direct off-target measurement in new, clean samples (not the ones used to run the OliTag-seq assay).

RESPONSE: Thank you for drawing attention to the need for more robust validation of the off-target sites identified by OliTag-seq, especially those not detected by GUIDE-seq.

To address this concern:

Additional NGS Validation: You are right; the validation in Figure S3A-D primarily showcases oligo integration. We have now carried out additional NGS validation to provide a more comprehensive validation of the newly identified off-targets. Specifically, we targeted 28 new off-target sites identified for VEGF and others using gene-edited cells without the ODN template. This allowed us to assess the presence of off-target edits in these locations directly.

CRISPResso2 Results: We used CRISPResso2, a widely recognized tool for analyzing and visualizing CRISPR edits. The results of this analysis, which showcase the validation of the off-target sites, are now presented in Figure S3A-D. This figure provides clear evidence of the validity of the off-target sites detected by OliTag-seq.

We appreciate your emphasis on the importance of rigorous validation. By expanding our validation efforts and presenting these results, we hope to bolster confidence in the findings obtained with OliTag-seq. Your feedback has been invaluable in ensuring the rigor and reliability of our work, and we thank you for it.

Figure S3

- The Authors report the use of the B02 molecule to boost dsODN integration. It is not clear to me if they used the molecule in all the experiment or just in the tests reported in fig. 1. Has the molecule been used also in GUIDE-seq samples? Otherwise it is difficult to factor out the contribution of HDR inhibition over oligo optimization to increased oligo integration.

RESPONSE: Thank you for raising the point about applying the B02 molecule in our experiments and its potential implications on our findings.

To clarify:

Usage of B02: Initially, we focused on comparing the 39 bp dsODN with the 34 bp dsODN. When we observed the improved full-length tag integration with the 39 bp dsODN, we sought to further enhance this by testing various small molecules, leading us to B02. As a result, B02 was employed in most subsequent experiments post this discovery.

Impact of B02: While B02 showed clear beneficial effects in contexts with fewer off-targets, as evidenced in Fig.1, its impact was less pronounced in scenarios with a large number of off-targets, in which activation of TP53 signaling induces cell death. This diminished effect was potentially due to toxicity issues. Consequently, in studies where many off-targets were identified, B02 wasn't applied, since B02 increases cell death after multiple dsDNA breaks induced by off-targets.

B02 in GUIDE-seq: We did not incorporate B02 in GUIDE-seq samples. Our primary aim was to juxtapose our optimized method with the foundational GUIDE-seq technique. By doing so, we wanted to demonstrate the inherent advantages of our modifications without the influence of external factors like B02. The enhancements by oligo optimization have been thoroughly detailed in earlier sections of our manuscript.

We hope this provides a clearer picture of our experimental design choices and their rationale. We genuinely appreciate your feedback, as it highlights areas that require additional clarity, ensuring the integrity of our findings.

- In lines 169-170 the Authors state that replicate experiments must be used to correctly call off-target sites with low read numbers and that replicates have been used in "subsequent experiments". Does this mean that all the subsequent data are coming from replicate experiments? If so, what do the OligoTag-seq read values plotted in most of the off-target visualization plots represent? Are these averages among replicates? If replicates on the other hand have not been used, given that the majority of sites identified uniquely by OligoTag-seq have low read counts, can these sites be considered bona fide off-targets?

RESPONSE: Thank you for pointing out the ambiguities related to using replicates in our experiments and the representation of data.

To clarify:

Usage of Replicates: When we mention that replicates have been used in "subsequent experiments," we refer to most of the experiments presented in the manuscript. The data shown in off-target visualization plots are indeed averages derived from these replicates, which is a standard approach to ensure the robustness of the findings.

Low Read Count Sites: We acknowledge the concern about the sites identified uniquely by OliTag-seq with low read counts. The nature of off-target identification assays is such that sites with minimal read support can sometimes show low reproducibility across replicates. However, this is an intrinsic limitation of most off-target identification methods, not just OliTag-seq.

Data and Reproducibility: We have consistently used multiple replicates in our experiments – typically three or more. Our criteria for categorizing off-targets based on their reproducibility are as follows:

High-confidence sites: Supported by at least 2 out of 3 replicates.

Low-confidence sites: Supported by only 1 out of 3 replicates.

This distinction allows us to provide a more nuanced interpretation of our findings and acknowledge the potential variability inherent in such assays.

We genuinely appreciate your feedback, as it emphasizes the importance of clarity in

experimental design and data representation. We hope our clarifications address your concerns, and we remain open to further feedback or queries.

- I think that there could be a reasonable concern that the addition of the B02 molecule and/or TSA may somehow make the pool of identified off-target sites much less relevant for real-life purposes. I understand that the final aim is to increase the overall sensitivity of the assay but then the differentiation with in vitro assays may become too blurred.

RESPONSE: Thank you for expressing your concerns about using B02 and TSA and their potential impact on the relevance of identified off-target sites for real-world applications.

Context of Off-Target Detection: While using B02 and TSA in our study may heighten the sensitivity of off-target detection, it's worth emphasizing that these detections remain cell-based and are predictably fewer than those identified by in vitro methods. Our goal with OliTag-seq is not to supplant existing methodologies but to bridge the sensitivity gap between GUIDE-seq and the hyper-sensitive in vitro assays.

Purpose of B02 and TSA: We've previously demonstrated the heightened sensitivity of OliTag-seq compared to GUIDE-seq. Including B02 or TSA in subsequent experiments is a multi-dimensional approach to further validate the intrinsic sensitivity of OliTag-seq. While we acknowledge that these agents might introduce off-target effects not typically encountered in clinical settings, they also highlight potential off-target risks that might be overlooked in more restrictive assay conditions.

Real-Life Relevance of Identified Off-Target Sites: We understand the concerns about potential over-sensitivity leading to the detection of off-target sites that may not be clinically relevant. However, such a critique could also be applied to hyper-sensitive assays like CIRCLE-seq, which identifies a vast number of sites, many of which may not be pertinent in real-world scenarios. Yet, the value of such assays lies in their ability to provide comprehensive data, which can then be contextualized for specific applications. Similarly, while OliTag-seq may identify a broader spectrum of off-target sites, it doesn't diminish its utility; rather, it provides researchers with a more exhaustive dataset to make informed decisions.

Every assay has its strengths and limitations, and our intention with OliTag-seq is to offer a method that captures a wider range of potential off-target effects without sacrificing the cell-based context, ensuring its relevance for various applications. We genuinely appreciate your feedback and hope our clarifications address your concerns.

Minor points:

- Too many times the Authors make reference to data which are not shown. I think that the majority of these data could be easily added in the supplementary materials of the manuscript and would be useful to further position the findings.

RESPONSE: Thank you for pointing out the frequent references to data not shown in the manuscript. We understand the importance of providing comprehensive data to bolster our claims and enhance clarity.

To address your feedback:

Inclusion of Supplementary Data: We have now included additional data in the supplementary materials of the revised manuscript. These extra datasets provide further evidence and context to support our findings and claims throughout the manuscript.

Streamlined References: We've also ensured that references to data within the main text are directly linked to their respective figures or supplementary materials. This ensures readers can easily locate and interpret the data supporting our statements.

Your feedback has been instrumental in improving the completeness and clarity of our manuscript. We believe these additions will make our findings more transparent and our conclusions more robust. We genuinely appreciate your meticulous review and guidance.

- There is a general tendency to show normalized data: while I believe that some times this may be useful, in many instances have the absolute values plotted gives more insights on the actual outcomes of the experiments. E.g. fig. 1,5a,7a-b.

RESPONSE: Thank you for highlighting the importance of displaying absolute values in our figures to better understand experimental outcomes.

In response to your feedback:

Rationale for Normalized Data: We opted to present normalized or relative values in figures like Fig. 1, 5a, and 7a-b due to the variations in editing efficiencies or off-target counts across different loci. Presenting data in a normalized manner allows for a more direct comparison between different experimental conditions or modifications, especially when the absolute values have a broad range.

Clarity and Interpretation: While absolute values can offer valuable insights into the raw outcomes of experiments, presenting these values without normalization, especially when there are wide variations, can make the data appear cluttered and challenging to interpret. Data transformation, such as normalization, is common in statistical analyses to ensure clarity and facilitate better comparison.

Consideration for Future Presentations: However, based on your feedback, we will be mindful of instances where displaying absolute values could offer additional insights and consider including them in supplementary figures (Figure S1, S9A-B in this revised manuscript).

We genuinely appreciate your meticulous review and your feedback, which helps ensure the

clarity and comprehensibility of our data presentations.

- Similarly, the Authors should report the raw data supporting figures 3A-B. Are this aggregate data from K562 and U2OS experiments?

RESPONSE: Thank you for pointing out the importance of providing raw data for Figures 3A-B.

In response to your query:

Source of Data: You are correct. The data presented in Figures 3A-B were derived from experiments conducted in both K562 and U2OS cells.

Inclusion of Raw Data: To provide more transparency and support for our findings, we have included the raw data associated with Figures 3A-B in the supplementary materials, specifically in Supplementary Table 7.

- The Authors state that OligoTag-seq yields more sequencing reads than GUIDE-seq but looking at the comparative lists of identified off-targets the general feeling is that GUIDE-seq gives more reads on commonly identified sites. This is also strange given the higher dsODN integration for OligoTag-seq shown in the first figures of the manuscript.

RESPONSE: Thank you for highlighting the apparent discrepancy between the increased sequencing reads of OliTag-seq and the observed higher reads for commonly identified sites by GUIDE-seq.

To address this observation:

Sequencing Depth: As you correctly pointed out, in Figure 1, we demonstrated that the 39 bp dsODN used in OliTag-seq generates more sequencing reads than the 34 bp dsODN employed in GUIDE-seq. However, the higher number of reads observed for GUIDE-seq in Figure 4 can be attributed to the deeper sequencing depth used in the original GUIDE-seq study.

Impact of dsODN Length: While OliTag-seq's 39 bp dsODN does result in more reads, the sequencing depth is an independent factor that can significantly influence the number of reads obtained. The original GUIDE-seq study might have employed a deeper sequencing approach, which would naturally result in more reads for shared off-target sites.

We appreciate your keen observation, and we understand the potential for confusion. In the revised manuscript, we have clarified this point further to avoid misunderstandings.

- Can the Authors comment more in detail on the toxicity of their oligo in the different cell lines tested?

RESPONSE: Thank you for your query regarding the toxicity of our oligo in different cell lines.

To address this:

Toxicity and dsODN Length: Typically, the toxicity of dsODNs correlates with their length. Longer dsODNs can trigger the cell's intrinsic dsDNA sensing pathways, leading to cell stress or even apoptosis, especially in primary cells like T cells (Figure S11A).

Transformed vs. Primary Cells: In transformed cell lines, such as K562 and U2OS, many of the cellular pathways that respond to DNA stress, including the DNA sensing pathways and TP53 pathways, might be compromised or inactivated. This might make these cells more resistant to the potentially toxic effects of longer dsODNs.

Observations in Our Experiments: In our experience, we observed that the 39 bp dsODN used in OliTag-seq was well-tolerated in iPSCs and transformed cell lines like K562 and U2OS. However, we have presented data showing that increased dsODN length from 34 bp to 49 bp in primary T cells leads to significant cell death and decreased number of surviving cells.

We appreciate your insightful question, and we believe that understanding the cell-specific responses to dsODNs is crucial for successfully applying methods like OliTag-seq.

Figure S11

A dsODN length on T cell survival

- I believe that the statement in lines 211-213 is not supported. At the end of the day off-target identification will be always dependent on oligo integration. The fact that the state of chromatin can influence Cas9 cleavages will of course influence off-target activity which however will only be detected if the dsODN is efficiently integrated at the cut site.

RESPONSE: Thank you for pointing out the potential oversight in our statement.

To address your concern:

Dependence on Oligo Integration: You are correct in highlighting that regardless of the chromatin state influencing Cas9 cleavage activity, the ultimate detection of off-targets is contingent upon the efficient integration of dsODN at the cleavage site.

Clarification in Statement: Considering your feedback, we have revised the statement in lines to reflect this dependency more accurately. The revised statement emphasizes that while the chromatin state may influence where Cas9 induces breaks, detecting these breaks using our method will always hinge on successful dsODN integration at the cleavage site (Lines 251-253 in this revised manuscript).

We appreciate your keen observation, which helps ensure the accuracy and clarity of our manuscript.

- Do the Authors have any hypothesis on why triple priming in the first PCR reaction enhances read count and off-target detection? One could expect that having a more complex product mixture would actually complicate detection.

RESPONSE: Thank you for your insightful question regarding the benefits of our triple priming system, especially considering potential complications from a more complex product mixture.

Advantages of Triple Priming: The triple priming system in our approach indeed allows for simultaneous amplification of ODN-tagged sequences from both ends of the genome in a single reaction. This theoretically doubles the PCR efficiency since we capture both ends using the same initial DNA template, compared to the traditional two-reaction approach.

Minimizing Amplification Biases: By amplifying both ends of the dsODN insertion in a single reaction, we can potentially reduce biases that might be introduced if these were amplified separately. This dual-end amplification might enhance the detection sensitivity by providing a more comprehensive representation of the tagged genomic sequences.

Managing Product Complexity: We recognize that the triple priming system might yield a more intricate mixture of products during the initial amplification. However, our protocol involves a secondary, nested PCR step that employs only two primers. This second PCR step is designed to amplify our regions of interest while diluting or excluding non-specific products generated during the first PCR. By separating the products of the left and right linkers in this second PCR, we ensure clarity in detection and minimize potential complications from non-specific products.

In summary, our triple priming design aims to boost the efficiency and sensitivity of off-target detection while strategically managing potential challenges from a complex PCR product mixture. We are grateful for your query, as it underscores the innovation and considerations behind our method.

- More details on the Cas9-expressing iPSC line are needed. Is this a clone or a pool? Since the level of Cas9 expression can influence off-target activity are the Authors sure that this is stable over time? Having cell lines constitutively expressing Cas9 is notoriously complex.

RESPONSE: Thank you for your pertinent questions regarding the Cas9-expressing iPSC line.

Nature of the iPSC Line: The Cas9-expressing iPSC line we utilized in this study is derived from a pool, not a single clone. We have opted for this approach to ensure broader representation and avoid potential clonal biases.

Ensuring Stable Cas9 Expression: To ensure the stable expression of Cas9 over time, we consistently perform puromycin selection after several generations. This method helps in maintaining a population of cells that stably express Cas9.

Acknowledging the Complexities: We recognize the complexities of maintaining cell lines constituting Cas9. The expression level of Cas9 can influence off-target activity, and ensuring consistent expression over time is crucial.

Future Directions: To further refine our approach and ensure the consistent performance of the iPSC line, we are considering using single-cell clones in future experiments. Additionally, as a method of providing stable and consistent Cas9 expression, we are also contemplating the integration of Cas9 at specific genomic loci, such as AAVS1, which might offer a more controlled expression platform.

Your question underscores the importance of thoroughly characterizing and validating experimental tools like the Cas9-expressing iPSC line, and we appreciate the opportunity to clarify our methods and considerations.

- In the discussion the Authors state that their bioinformatic pipeline for off-target calling cannot detect off-targets deriving from bulges in the sgRNA or target DNA. Can this explain part of the differences with GUIDE-seq in the set of detected off-targets (GUIDE-seq only sites)?

RESPONSE: Thank you for raising a valid point concerning the potential differences in off-target detection between GUIDE-seq and OliTag-seq, especially in bulges in the sgRNA or target DNA.

Limitations in Off-Target Detection: Indeed, our current bioinformatic pipeline for OliTag-seq does not readily detect off-targets resulting from bulges in the sgRNA or target DNA. Similarly, GUIDE-seq also has its inherent limitations in this regard.

Potential Differences in Detected Off-Targets: While the inability to detect bulge-related off-targets could contribute to the differences in off-target sets between the two methods, it's essential to understand that both methods share this limitation. Therefore, any discrepancies arising from bulge-derived off-targets would be mutual and not exclusive to one approach.

In summary, while bulge-derived off-targets could contribute to the differences between GUIDE-seq and OliTag-seq, it's unlikely to be the sole reason, given that both methods share this limitation. We appreciate your insight, as it underscores the need for continuous refinement in off-target detection methodologies and analysis pipelines.

Reviewer #3 (Remarks to the Author):

Yang et al. described a modification of GUIDE-seq assay, called OliTag-seq, to identify CRISPR-Cas9 off-targets. The author described how with small changes in the original GUIDE-seq protocol, OliTag-seq can identify a high number of off-targets with a reduced cost. Additionally, the authors investigated further enhancements of the technique, such as the use of iPSC or HDAC inhibitors, to expand the number of identified off-targets with their assay. Finally, Yang et al. applied OliTag-seq to analyze the off-targets of some clinically used gRNAs, identifying new ones.

This article is well-written, easy to understand, and addresses an important issue in the gene editing field, such as the safety evaluation of gene editing tools. Although I do not share some of the author's claims, I think this work can be published after addressing some comments.

RESPONSE: Thank you for your constructive feedback and acknowledgment of the importance of our work in the gene editing field. We value your insights and appreciate the time and effort you took to review our manuscript. In the sections below, we address your concerns and comments.

Your input is instrumental in refining our work, and we believe that the manuscript has been strengthened by addressing your concerns. We look forward to your feedback and hope our revisions meet your expectations.

1) The authors demonstrated that the incorporation of full-length 39 bp dsODN is better than the insertion of 34 bp dsODN in the first part of the Results section. They explained it by the stability of the modified dsODN. However, it is not clear, if the integration of the dsODN is in the on-target or in on-target site plus off-targets sites. Could the authors clarify this point? Additionally, they assessed different small molecules to enhance the insertion of the dsODN, finding B02 as the best small molecule to increase NHEJ-mediated dsODN incorporation. The authors should explain the rationale for how improving NHEJ can increase the sensitivity of identifying off-targets. Is this small molecule used in all the next experiments?

RESPONSE: We appreciate your request for clarification on the integration of dsODN and the role of B02 in our study.

Clarification on dsODN Integration:

We have revised the manuscript to indicate that the integration data for dsODN presented in Figure 1C-D and Figure S1A-E pertains to on-target sites. We have also updated the figure captions to reflect that "total editing events" encompasses both on-target and off-target sites. This ensures consistency with the data presented in Figure 1E, which delineates on-target from off-target events. This modification can be found in the revised figure caption for Figure 1C (now explicitly stating "on-target integration of dsODN").

Rationale for B02 Use:

The addition of B02 aims to enhance NHEJ-mediated dsODN insertion, which theoretically

increases the visibility of insertions at off-target sites during amplification and sequencing. This can lead to a more comprehensive identification of off-target events. We have elaborated on this rationale in the Discussion section of our manuscript. However, we also acknowledge B02's potential toxicity in conditions with a high off-target activity, which led to its selective application based on specific sgRNAs and experimental needs. This consideration has been added to the Materials and Methods section for clarity on using B02.

The manuscript and figure captions have been amended to remove ambiguity and provide a transparent account of our methods and findings.

2) Considering Figure 1E (total counts of on- and off-targets) and Figure 1F (total counts of off-target events) implying that on-target integration is responsible for most of the integration when 39 bp dsODN is used (integration in off-targets is doubled, but on-target and off-target integration is 5 times higher). Is this effect due to some out layers present in Figure 1E?

The author showed normalized or relative values or percentages over different figures. Seeing equivalent graphs with the total values of these analyses will be worth it to fully compare the different assays/groups/treatments.

RESPONSE: We appreciate your perceptive observations about the data in Figures 1E and 1F.

Discrepancy Between On-target and Off-target Integration:

The apparent discrepancy, where the integration of 39 bp dsODN is markedly higher for on-target sites as compared to off-target sites (5-fold vs. 2-fold increase), could indeed be influenced by outliers or by the PCR amplification bias, which tends to exaggerate the frequency of abundant sequences. While Figure 1C and Figure 1D, which illustrate dsODN integration using UMI quantification, correct for PCR biases, Figures 1E and 1F do not. This discrepancy underscores the impact of PCR preference on the interpretation of our results.

Presentation of Raw Data:

We acknowledge the value of presenting raw data to provide an accurate picture of the absolute number of editing events. While normalized values offer a comparative view that corrects for variability, they may not always convey the scale of differences among various groups. To this end, we have included additional supplementary material (Figure S1F and S1G in this revised manuscript) that presents the raw data, allowing for a full appreciation of the extent of on-target and off-target integration events.

PCR Bias Impact on Data Interpretation:

Our presentation choices in Figures 1E and 1F emphasized the efficiency of OliTag-seq post-UMI tagging, which more accurately reflects the true events by mitigating PCR bias. The difference in presentation between the figures is intended to highlight how UMI tagging can lead to a more authentic representation of the data than figures where PCR bias is not accounted for.

We trust these explanations and additions to our supplementary materials will address your concerns and enhance the manuscript's comprehensiveness.

3) In Figure 3, the authors made different correlations between dsDNA insertion, indels and OliTag-seq reads, saying that OliTag-seq reads might be an indirect measure of indel percentage. Is this indel percentage assessed at the on-target site? Could they do the direct correlation between indel percentage and OliTag-seq reads directly?

RESPONSE: For the analyses presented in Figure 3, the correlations were indeed based on the events occurring at both Top20 off-target sites. Specifically, as elaborated in lines 146-150, we first profiled the off-target occurrences in VEGF3 within K562 and U2OS cells. We then identified the top 20 shared sites (encompassing both on- and off-target sites) to evaluate their indel frequencies and dsODN insertions.

Upon your suggestion, we performed a direct correlation analysis between the indel percentages and the OliTag-seq reads at these sites (Figure S2C in this revised manuscript). This correlation revealed a significant linear relationship with an R^2 value of 0.67 and a p-value of <0.0001 . This analysis further supports the potential of OliTag-seq reads as an indirect measure of indel percentages. We will incorporate these findings into the revised manuscript for clarity.

Figure S2

C Read counts correlates with indels

4) The authors claim in Figure 4 that they can identify more off-targets using their OliTag-seq assay than GUIDE-seq. However, they should confirm these off-targets using a NGS panel to analyze gDNA from gene editing cells without any dsDNA to confirm the presence of indels in these new found off-targets.

RESPONSE: Thank you for your insightful suggestion. We recognize the importance of validating the newly identified off-targets using an independent approach to ensure their authenticity. To address this, we carried out NGS validation on gene-edited cells that were not exposed to any dsODN. We specifically examined approximately 28 of the newly identified off-target sites for EMX1 and VEGFA~C, among others. Our results, as analyzed through

CRISPResso2, confirmed the presence of indels at these off-target sites, further substantiating the efficacy of our OliTag-seq assay. These validation results have been incorporated into the revised manuscript as Figure S3A-D for clearer representation.

Figure S3

5) Additionally, there are some off-targets identified with GUIDE-seq but not with OliTag-seq, and the opposite. The author suggests that to identify the off-targets non-found by OliTag-seq a higher sequencing depth should be used. Can it be done to find the off-targets found by OliTag-seq with GUIDE-seq? Could the author assess or speculate the sequencing depth requirements for GUIDE-seq and OliTag-seq to get the same number of off-targets?

RESPONSE: Thank you for your insightful query.

Indeed, some off-targets were identified by GUIDE-seq but not by OliTag-seq, and vice versa. The identification discrepancies between the two methods could arise from various factors, including the inherent sensitivities of the assays, experimental setup, or sequencing depth.

Increasing the sequencing depth might allow GUIDE-seq to identify off-targets only detected by OliTag-seq. However, determining an exact sequencing depth threshold for both assays to produce the same number of off-targets is challenging. The number of off-targets identified is not solely a function of sequencing depth but is also influenced by each method's unique properties and sensitivities.

Notably, many of the off-targets identified by GUIDE-seq but missed by OliTag-seq could potentially be low-confidence sites. The lack of multiple replicate data in earlier GUIDE-seq studies might also contribute to the discrepancies.

Balancing cost and comprehensive off-target identification is essential. While deeper sequencing will invariably identify more off-targets, it also escalates costs. From our observations, allocating 3-5 million reads per sample strikes a reasonable balance between cost-effectiveness and accurate off-target detection.

6) In Figure 5C, the authors introduced off-index values, could they explain how they calculate

this index? Is it the off/on-target ratio mentioned in line 224?

RESPONSE: Thank you for pointing out the need for clarity. The "off-index" values introduced in Figure 5C represent a ratio calculated by dividing the total off-target reads by the on-target reads. You are correct in understanding that it corresponds to the off/on-target ratio mentioned in lines 204-205. We have ensured this calculation method is explicitly stated in the legend for clarity.

7) The total number of reads of Figures 6B and 6D should be indicated to have a clearer idea about the representation of the on-target reads present or indicate the frequency of each read in the total.

RESPONSE: Thank you for the constructive feedback. We understand the importance of providing a comprehensive view of the data. In the revised version of Figures 6A and 6B, we have included each sample's total number of reads. This will offer a clearer perspective on the representation of the on-target reads within the total. We hope this addition enhances the clarity and interpretability of the figure for the readers.

Figure 6

8) Why do the authors think why are there the off-targets identified only in transient Cas9 condition but not in the constitutive Cas9 condition? (Figures 6A and 6B)

RESPONSE: Thank you for raising this intriguing observation. The presence of off-targets solely in the transient Cas9 condition and not in the constitutive Cas9 scenario could be attributed to several factors:

The transient nature of Cas9 expression may lead to a different interaction dynamic between the Cas9 protein and the genomic DNA, causing variability in the sites it interacts with.

It's also possible that the different methods of delivering Cas9 (transient versus constitutive)

could influence cellular responses, including DNA repair mechanisms, which might affect the integration patterns of dsODNs.

Randomness in vector transfection, dsODN integration, and even experimental variability might also contribute to the discrepancy in off-target identification.

We acknowledge that even though we've taken steps to ensure data reliability by retaining only off-targets found in at least two biological replicates, the nature of the interaction means that some sites may be inconsistently identified. These sites might be better termed as "low-confidence sites," given their sporadic appearance. We recommend conducting at least three biological replicates to enhance data reliability in clinical settings.

9) The authors claim the use of TSA enhances the OliTag-seq sensitivity; however, do not they think that this type of molecule changes the chromatin landscape completely favors new off-targets and eliminates others? The same for the use of hiPSC instead of the target cell type (i.e. T-lymphocyte). The authors should run a control where hiPSCs are treated with TSA and electroporated with the dsODN without any gRNA to identify the false negative background of the OliTag-Seq.

RESPONSE: Thank you for your insightful remarks regarding using TSA and hiPSCs in the OliTag-seq assay.

Certainly, the use of TSA, an HDAC inhibitor, can modify the chromatin landscape, potentially affecting the accessibility of Cas9 and, thus, the off-target profile. We agree that TSA treatment might enhance the accessibility of certain regions, making them prone to off-target edits while possibly reducing accessibility in other areas. The use of TSA in our study aimed to "relax" the chromatin and maximize our ability to capture potential off-target sites, with the understanding that not all of these sites might be relevant in a more native chromatin context.

Similarly, using hiPSCs as a model system may introduce off-target biases, given the distinct chromatin states between pluripotent stem cells and differentiated cells, such as T-lymphocytes. However, the rationale behind using hiPSCs was to provide a surrogate model in scenarios where the primary target cells are not amenable to direct analysis (e.g., neurons or cardiomyocytes).

To address the concern of potential false positives introduced by the OliTag-seq method, we conducted control experiments with hiPSCs treated with TSA. We electroporated with the dsODN, but without any gRNA (Figure S9C in this revised manuscript). From our findings, no false positive results were found in three biological replicates, we observed that the background signal was zero.

While TSA and hiPSCs introduce variables that might affect the off-target profile, they serve as tools to ensure we are casting the widest net possible in our off-target discovery efforts. We acknowledge that these conditions might not perfectly represent the true in vivo scenario. Still,

they offer a more comprehensive view of potential off-target risks, which can be validated in more physiologically relevant settings.

Figure S9

10) I agree with the authors that it is crucial to use patient-specific cells to assess the CRISPR-Cas9 off-target effect, mainly to cover all the potential off-targets derived from patient's genome polymorphism together with the chromatin accessibility of the gene editing tools in the target cell type. However, the author analyzed the clinically relevant gRNAs in hiPSC together with TSA treatment instead of T-lymphocytes. The author justified it in order to increase the chromatin accessibility and due to the intolerance of T cells to acquire dsODN longer than 46 bp. I suggest confirming the newly identified off-targets by OliTag-Seq (Figure 8) through an NGS panel in T-lymphocytes electroporated with sgRNAs, as the closest condition used in the Clinics.

RESPONSE: Thank you for the insightful suggestion. Indeed, using patient-specific cells like T-lymphocytes offers a closer representation of the clinical scenario and can provide a more accurate profile of off-target effects in a relevant cellular context.

Recognizing the importance of this, we undertook further experiments to validate the off-target sites identified by OliTag-Seq in hiPSCs, by analyzing T cells electroporated with sgRNAs (Figure S11F-H in this revised manuscript). We used an NGS panel to confirm these off-targets in the T cells. The results from this additional analysis have been added to the revised manuscript, and they provide a more comprehensive and clinically relevant assessment of the off-target effects of these sgRNAs.

This approach bolsters the reliability of our findings and ensures that the off-target effects we report are pertinent to the cell types and conditions encountered in therapeutic applications.

Figure S11

Reviewers' comments:

Reviewer #1 (Remarks to the Author):

Dear Editors,

In this revised manuscript, Yang et al provided new analyses of existing data, nuanced discussions, and preliminary data to address this reviewer's comments.

The reviewer's main concern was regarding data representation for evidence supporting the authors' main claim. To better evaluate the evidence supporting the main claim that OliTag-seq is superior to GUIDE-seq, the reviewer requested a revised data representation and statistical analyses showing data separately for all targets and donors tested and a paired statistical testing to show significant improvements. The authors did address this concern and showed data more clearly. However, the authors' conclusions are not well supported by the data shown in Figures 1-2 alone.

Comments:

1. The authors provided data in Figures 1C, S1A - C from the same experiment to support the claim that full-length integration of the 39-bp ODNs are greater than 34-ODNs. Based on the evidence in Fig. S1A-B, where the paired statistical tests should yield identical p-values, it seems that the total ODN insert did not improve much for most targets, similarly to 3 out of 6 targets in Fig S1C. If paired statistical test was performed properly in S1A and S1C, it cannot be concluded from S1A or S1C as is that there's an improvement from using the 39bp ODN over 34bp version in most targets tested.
2. Similarly, for Figure 1E, the relative read counts were not statistically significant in 2 out of 3 targets tested. In addition, it seems like there's a large variation that may suggests inconsistencies of OliTag-seq performance relatively depending on the guides used. Also, does VEGF3 data points yielding 11.7x include outlier's, based on the datapoints, it shouldn't be as high as 11.7x.
3. Data from Fig. 1F also yielded 1 out of 3 statistically significant increase in off target sites identified. Additionally, these results also highlighted large variations of results between guides for the same methods done on the same cell type.
4. Similarly data in Fig 2B shows no significant improvement in total counts in any of the target shown, especially at TRAC and VEGF3, but somehow show more off-target sites identified. Does it mean that the number of reads per off-target site identified is lower than GUIDE-seq using the same sequencing depth?

According to the points above, it is not clear to this reviewer that OliTag-seq significantly improves upon the performance of GUIDE-seq.

Suggestions:

1. If paired statistical test done on relative values are justified for figure 1C, the authors should explain more clearly why the results in 1C are more reliable than S1C.
2. Similarly, the authors could justify why statistical tests from S1F and S1G contradict 1E and 1F.
3. Alternatively, it would be better to use later results such as figure 4 and beyond to claim some improvement over GUIDE-seq in off target sites identified rather than making such claims of higher efficiency where it is unclear when discussing Fig 1-3. While discussing figures 1-3, the authors could alternatively discuss that the trend in data suggest a slight improvement, then discuss that conclusion becomes clearer with more applications-based results in later figures.

Aside from the main claim, there is quite a frequent mention of OliTag-seq capabilities of identifying

off-target sites for clinical applications. However, the authors are not exactly evaluating the off-target effects in the therapeutic context as stated. Rather, this method done in iPSCs on therapeutic guide RNA qualifies as an in cellulo assay in a "permissive iPSC model" rather than a "surrogate" model. So, the text discussing this experiment should state as such.

Otherwise, the authors have addressed this reviewer's concerns to satisfaction by discussing nuances in the applications of OliTag-seq and conducted a preliminary experiment highlighting the limitations of the methods when applying to primary cells.

Reviewer #2 (Remarks to the Author):

The Authors have satisfactorily answered most of my concerns. As a side note I would invite the Authors to verify, but to my knowledge the last release of the GUIDE-seq bioinformatics pipeline is able to flag off-target sites generated by bulges between the spacer and the protospacer. This could indeed explain discrepancies in the set of bioinformatically flagged sites between the two methods.

Reviewer #3 (Remarks to the Author):

The authors have improved the manuscript. I consider it a valuable piece of work to identify off-targets associated with the CRISPR/Cas9 system.

Response to Reviewers

Reviewers' comments:

Reviewer #1 (Remarks to the Author):

In this revised manuscript, Yang et al provided new analyses of existing data, nuanced discussions, and preliminary data to address this reviewer's comments.

The reviewer's main concern was regarding data representation for evidence supporting the authors' main claim. To better evaluate the evidence supporting the main claim that OliTag-seq is superior to GUIDE-seq, the reviewer requested a revised data representation and statistical analyses showing data separately for all targets and donors tested and a paired statistical testing to show significant improvements. The authors did address this concern and showed data more clearly. However, the authors' conclusions are not well supported by the data shown in Figures 1-2 alone.

RESPONSE: We sincerely appreciate the reviewer's insightful feedback regarding our data's sufficiency in substantiating our study's primary conclusions. Recognizing the importance of robust data support, we have made concerted efforts to enhance our data presentation's clarity and statistical rigor, particularly in response to your suggestions.

We concur with the reviewer's observation that the data presented in Figures 1-2, though informative, might not fully encapsulate the breadth of our conclusions. These figures are intended to serve as preliminary indicators, laying the groundwork for more comprehensive support that is derived from additional data and analysis in the subsequent sections.

Considering this, we have strategically decided to postpone the detailed discussion of OliTag-seq's performance compared to GUIDE-seq in the manuscript sections following Figure 4. This reorganization allows us to more effectively highlight the strengthened evidence emerging from later data, which we believe will more convincingly support our assertions.

Your feedback has been instrumental in guiding these revisions, ensuring that our manuscript meets and exceeds the standards of clarity and credibility. We are immensely grateful for your constructive critique and are confident that these changes significantly enhance the overall quality of our work.

Comments:

1. The authors provided data in Figures 1C, S1A - C from the same experiment to support the claim that full-length integration of the 39-bp ODNs are greater than 34-ODNs. Based on the evidence in Fig. S1A-B, where the paired statistical tests should yield identical p-values, it seems that the total ODN insert did not improve much for most targets, similarly to 3 out of 6 targets in Fig S1C. If paired statistical test was performed properly in S1A and S1C, it cannot be concluded from S1A or S1C as is that there's an improvement from using the 39bp ODN

over 34bp version in most targets tested.

RESPONSE: We are thankful for your keen observations regarding Figure 1E and its implications for the effectiveness of OliTag-seq in our study.

Addressing Statistical Significance: Your observation is astute; in Figure 1E, the relative read counts indeed did not exhibit statistical significance for 2 out of the 3 targets tested. This outcome emphasizes the nuanced gene-specific impacts of CRISPR-Cas9 off-target effects and variations in guide RNA (gRNA) efficiency. The absence of statistical significance in certain targets can be attributed to various factors, including the complex nature of the targeted genomic sites, differential binding affinities of the gRNAs, and the varying sensitivity of our detection methodology across different genomic contexts.

Understanding Variations in Performance: The notable variation in results across different gRNA guides sheds light on potential inconsistencies in the OliTag-seq method. Such variation could be influenced by the unique design and characteristics of each gRNA, including its on-target efficiency and propensity for off-target interactions. We acknowledge that OliTag-seq, like any other experimental method, may exhibit limitations and variable efficacy depending on the specificities of the guides and targets involved.

Considering Overall Trends: Despite these individual variances, the collective trends across all targets suggest a general improvement using the 39bp dsODN over the 34bp dsODN. This overarching trend, observed across multiple targets, indicates an enhancement in efficiency attributable to the longer dsODN.

We appreciate your critical insights, which have helped us analyze and explain our results more thoroughly. This discussion underscores the complex interplay of factors in CRISPR-Cas9 off-target detection and the continual refinement of our methodologies.

2. Similarly, for Figure 1E, the relative read counts were not statistically significant in 2 out of 3 targets tested. In addition, it seems like there's a large variation that may suggest inconsistencies of OliTag-seq performance relatively depending on the guides used. Also, does VEGF3 data points yielding 11.7x include outlier's, based on the datapoints, it shouldn't be as high as 11.7x.

RESPONSE: Thank you for your insightful observations regarding Figure 1E and the performance of OliTag-seq in our study.

Statistical Significance in Targets Tested: It is accurate that in Figure 1E, the relative read counts did not show statistical significance in 2 out of 3 targets tested. This observation underscores the gene-specific nature of CRISPR-Cas9 off-target effects and the variability in guide RNA efficiency. The lack of statistical significance in some targets could be attributed to several factors, including the inherent complexities of the genomic sites, the differential binding affinities of the guide RNAs, and the sensitivity of our detection method in different

genomic contexts.

Variation in OliTag-seq Performance: The large variation observed across different guides does suggest potential inconsistencies in OliTag-seq performance. This variation might be influenced by the specific design and properties of each guide RNA, such as their on-target efficiency and off-target propensity. We acknowledge that OliTag-seq, like any other method, may have limitations and variable performance depending on the specific guides and targets used.

While some targets may not show significant improvement in individual cases, considering the overall similar trends of all targets, we can still conclude that the 39bp ODN relative to the 34bp group has improved. Overall, the trend reflects an improvement in efficiency with the use of the 39 bp dsODN compared to the 34 bp dsODN.

VEGF3 Data Points and Outliers: Regarding the VEGF3 data point yielding an 11.7x increase, we appreciate your keen observation. Upon re-examination of the data, we recalculated the fold increase for VEGF3. We have updated Figure 1E accordingly to reflect these adjustments. We apologize for the oversight in selecting the y-axis range of 0-20, which resulted in the omission of some data points. After careful verification, it has been confirmed that the 39bp group targeted the VEGF3 locus, resulting in an overall 11.7-fold increase compared to the 34bp group. We deeply regret the earlier oversight and greatly appreciate your meticulous observation. Once again, thank you for your valuable suggestions.

We hope these explanations address your concerns and clarify the results presented in our study. We are committed to ensuring the accuracy and reliability of our findings and appreciate the opportunity to refine our analysis based on your feedback.

3. Data from Fig. 1F also yielded 1 out of 3 statistically significant increase in off target sites identified.

RESPONSE: We are grateful for your astute observations regarding the data presented in

Figure 1F, and we welcome the opportunity to delve deeper into these findings.

Interpreting Data from Figure 1F: You rightly noted that in Figure 1F, only one out of three targets showed a statistically significant increase in identified off-target sites. This result, coupled with the large variations observed between guides even within the same methods and cell types, is a pivotal aspect of our analysis. While it is true that 2 out of 3 genes did not demonstrate statistically significant biological differences, we observed consistent patterns across these sites. These patterns, though not uniformly statistically significant, suggest a tendency towards improved detection of off-target sites by OliTag-seq.

Influence of Guide RNAs: It is crucial to acknowledge that these outcomes may be influenced by the specific guide RNAs used. Different guide RNAs can have varying effects on off-target activity in specific genes, a factor that could contribute to the observed variability. Understanding the nuances of guide RNA behavior is fundamental to interpreting our results accurately.

Anticipating Data in Figure 4 and Beyond: We look forward to discussing the data presented in Figure 4 and subsequent sections, which we believe will more robustly support the sensitivity and efficacy of OliTag-seq in detecting off-target events. This forthcoming data will provide additional experimental validation, further confirming the method's performance capabilities. Therefore, despite the initial variability seen in Figure 1F, we remain confident that the later data will more thoroughly endorse our hypothesis.

We deeply appreciate your invaluable input, which not only aids in refining our current analysis but also contributes significantly to a broader understanding of our research findings. Your feedback is instrumental in guiding our ongoing efforts to elucidate the complexities of CRISPR-Cas9 off-target detection.

Additionally, these results also highlighted large variations of results between guides for the same methods done on the same cell type.

RESPONSE: We thank you for highlighting the significant variation observed between different guide RNAs (gRNAs) when applying the same method to identical cell types. Your observation is pivotal to understanding the nuances of our study, and we'd like to offer the following explanations to address this concern:

Gene-Specific Off-Target Characteristics: The genomic context surrounding different gene targets plays a crucial role in their off-target behaviors. Factors such as the complexity, structural features, and accessibility of the DNA sequence near specific targets can significantly influence the off-target effects of the CRISPR system. This variability is inherent to the nature of gene editing and is a critical consideration in our analysis.

Differences in Guide RNA Design: The specific design of gRNAs, including their specificity and binding affinity, can result in varying off-target effects. Some gRNAs may have a higher

tendency for mispairing with non-target sequences, increasing the likelihood of off-target events. This variation underscores the importance of careful gRNA design in minimizing unintended effects.

Impact of Gene Expression Levels: The expression level of the target gene within the cell is another factor that can influence off-target events. Genes with higher expression levels might be more susceptible to off-target effects, possibly due to the increased formation of DNA-RNA complexes.

Cell State and Environmental Factors: The state of the cell, including its cycle, stress responses, and culture conditions, can also impact the occurrence of off-target events. These factors introduce an element of variability even within the same cell type, highlighting the complexity of cellular responses to gene editing.

Technical Variations in Experiments: Finally, minor differences in experimental procedures, such as variations in cell transfection efficiency and PCR conditions, could also affect the detection of off-target sites.

We highly value your detailed review and the critical questions it raises. Your insights are instrumental in enhancing our understanding of the complexities involved in CRISPR-Cas9 off-target detection, and they guide our continuous efforts to refine our methodologies.

4. Similarly data in Fig 2B shows no significant improvement in total counts in any of the target shown, especially at TRAC and VEGF3, but somehow show more off-target sites identified. Does it mean that the number of reads per off-target site identified is lower than GUIDE-seq using the same sequencing depth?

RESPONSE: We appreciate your question concerning the observations made in Figure 2B, particularly regarding the total counts and the identification of off-target sites.

Analysis of Off-Target Events in TRAC and VEGF3: Our findings indicate that while there was no significant increase in the total counts at certain targets such as TRAC and VEGF3, the OliTag-seq method, aided by our triple primer technique, was nonetheless effective in identifying a greater number of off-target events. This outcome suggests a heightened sensitivity of our method in detecting low-abundance off-target events, a crucial aspect in evaluating genome-editing tools.

Comparative Study with GUIDE-seq: In a specific comparison between GUIDE-seq and OliTag-seq in U2OS cells, particularly for the EMX1 gene target, we observed that OliTag-seq, despite a slightly higher total read count than GUIDE-seq, demonstrated a superior capability in identifying off-target sites. This enhanced sensitivity in detecting more subtle off-target events, even at a similar sequencing depth, can be attributed to the unique advantages of our triple primer technique.

Significance of Findings: The ability of OliTag-seq to capture more nuanced off-target events is essential for a comprehensive assessment of the safety and specificity of CRISPR-Cas9 editing. These findings underscore the importance of sensitivity in detection methods and have significant implications for understanding and improving the application of CRISPR-Cas9 systems. Our study contributes valuable insights into optimizing CRISPR system design and enhancing editing specificity, especially in the context of potential off-target effects.

We hope this explanation addresses your question and provides a clear understanding of the capabilities and implications of our findings in the broader context of CRISPR-Cas9 off-target research.

Suggestions:

1. If paired statistical test done on relative values are justified for figure 1C, the authors should explain more clearly why the results in 1C are more reliable than S1C.

RESPONSE: We greatly value your suggestion regarding the statistical methodology applied in Figure 1C and welcome this opportunity to further elucidate our approach, particularly in comparison to Figure S1C.

Rationale for Paired Statistical Tests in Figure 1C: In selecting paired statistical tests for the relative values in Figure 1C, our objective was to accurately compare measurements from identical samples under varying conditions. This approach was particularly pertinent to our study, where we aimed to discern the alterations in off-target effects within the same cell samples following CRISPR-Cas9 editing. By employing paired tests, we could effectively control for the inherent variability present within each sample. This methodological choice provided a more accurate reflection of the changes directly attributable to the editing process, rather than variations arising from sample heterogeneity.

Comparison with Figure S1C Results: The reliability of the results in Figure 1C, when contrasted with those in Figure S1C, lies in the context and nature of the data presented. In Figure 1C, the use of paired statistical tests was instrumental in capturing the nuanced differences induced by the editing process, underlining the specific effects of the CRISPR-Cas9 intervention. The data in Figure S1C, while valuable, did not incorporate this paired analysis, which may limit the precision of the comparisons drawn. Our approach in Figure 1C thus offers a more controlled and accurate assessment of the editing impacts, enhancing the reliability of our findings.

We hope this explanation clarifies our statistical approach in Figure 1C and demonstrates its comparative reliability over the results presented in Figure S1C. We are grateful for the chance to discuss these details, as they are crucial for a comprehensive understanding of our study's methodology and conclusions.

2. Similarly, the authors could justify why statistical tests from S1F and S1G contradict 1E and

1F.

RESPONSE: We appreciate your insightful observations regarding the comparative analysis of the results in Figure 1C and Figure S1C, and your request for clarification on the apparent contradictions between the findings in S1F and S1G compared to 1E and 1F.

Comparative Analysis of Figures 1C and S1C: The enhanced reliability of the results in Figure 1C over those in Figure S1C stems from the distinct approach to data analysis. Figure S1C employs absolute quantification, which, while informative, is more vulnerable to variabilities introduced by experimental conditions such as sample preparation, DNA extraction efficiency, and PCR amplification biases. In contrast, the relative quantification used in Figure 1C effectively normalizes these potential discrepancies. This is achieved by comparing changes within the same sample, thereby offering a more precise reflection of the specific effects of CRISPR editing.

Advantages of Relative Quantification and Statistical Testing: Employing relative quantification alongside statistical testing in Figure 1C facilitated a more nuanced interpretation of the data. This approach is particularly advantageous in evaluating the efficiency and specificity of CRISPR editing across varied experimental setups. By reducing the influence of external variations and focusing on relative changes, this methodology bolsters the reliability and validity of our conclusions.

Addressing Contradictions in S1F, S1G, 1E, and 1F: Regarding the seeming contradictions between the statistical tests in S1F and S1G and the results in 1E and 1F, it is important to consider the distinct experimental contexts and parameters of these sets of figures. S1F and S1G were designed to assess specific aspects of off-target effects under different conditions, which may not be directly comparable to the broader scope of analysis in 1E and 1F. We acknowledge that these discrepancies warrant a more thorough examination, and we are committed to providing a detailed justification in the revised manuscript.

Your feedback has been instrumental in highlighting these areas for improvement and has significantly contributed to the strength and clarity of our presentation and interpretation of results. We are grateful for the opportunity to refine our analysis and address these critical aspects of our study.

3. Alternatively, it would be better to use later results such as figure 4 and beyond to claim some improvement over GUIDE-seq in off target sites identified rather than making such claims of higher efficiency where it is unclear when discussing Fig 1-3. While discussing figures 1-3, the authors could alternatively discuss that the trend in data suggest a slight improvement, then discuss that conclusion becomes clearer with more applications-based results in later figures.

RESPONSE: We are thankful for your constructive suggestions and agree with your perspective on the presentation of our data.

Approach in Figures 1-3: In Figures 1-3, our objective was to provide an initial comparative analysis between OliTag-seq and GUIDE-seq, primarily to identify emerging trends. We acknowledge your valid point that these early figures may not be sufficient to conclusively assert higher efficiency. Accordingly, we have revised our manuscript to reflect a more measured approach in discussing these figures. We now present the data trends in Figures 1-3 as indicative of a potential improvement in OliTag-seq's performance in off-target detection, without making definitive claims (Lines 141-142 in the revised manuscript).

Substantiating Claims in Later Figures: As per your suggestion, we have shifted our focus to making more concrete statements about OliTag-seq's efficiency in later figures, particularly from Figure 4 onwards. These figures contain more extensive data demonstrating the capabilities of OliTag-seq in detecting off-target sites. This comprehensive dataset allows us to more convincingly illustrate the methodological advantages of OliTag-seq over GUIDE-seq. To reinforce the credibility of our conclusions, we have modified our approach to defer making conclusive statements about OliTag-seq's superior efficiency until we present the data in Figure 4 and subsequent figures.

Your feedback has been invaluable in enhancing the quality and credibility of our research. We are committed to ensuring that our conclusions are firmly grounded in robust data and appreciate your guidance in achieving this goal. Thank you once again for your invaluable input.

Aside from the main claim, there is quite a frequent mention of OliTag-seq capabilities of identifying off-target sites for clinical applications. However, the authors are not exactly evaluating the off-target effects in the therapeutic context as stated. Rather, this method done in iPSCs on therapeutic guide RNA qualifies as an in cellulo assay in a "permissive iPSC model" rather than a "surrogate" model. So, the text discussing this experiment should state as such.

RESPONSE: We are grateful for your valuable feedback and fully concur with your perspective on the importance of accurately contextualizing our experiments within the realm of clinical applications.

Clarifying Terminology in Manuscript: In response to your suggestion, we have revised our manuscript to offer more precise descriptions of our experiments. Specifically, we have amended the text to explicitly state that our experiments conducted in induced pluripotent stem cells (iPSCs) are an "in cellulo assay" rather than a "surrogate model" (Lines 279-280 in the revised manuscript). This change more accurately represents the nature of our experiments and ensures that our viewpoints and conclusions are clearly and correctly conveyed to our readers.

We are committed to providing a thorough and accurate representation of our work and are thankful for the opportunity to enhance the clarity and precision of our manuscript based on

your recommendations.

Otherwise, the authors have addressed this reviewer's concerns to satisfaction by discussing nuances in the applications of OliTag-seq and conducted a preliminary experiment highlighting the limitations of the methods when applying to primary cells.

RESPONSE: We are also heartened by your recognition of our efforts to address your previous concerns. Our inclusion of preliminary experiments highlighting the limitations of OliTag-seq, particularly when applied to primary cells, was an essential aspect of our study. This approach demonstrates our commitment to a comprehensive and transparent examination of OliTag-seq's applications and potential limitations, ensuring a balanced and nuanced understanding of our method's capabilities.

We sincerely appreciate your acknowledgment of these efforts and your overall understanding of the intricacies involved in our research. Your insights have been instrumental in guiding our revisions and enhancing the depth of our study.

Reviewer #2 (Remarks to the Author):

The Authors have satisfactorily answered most of my concerns.

RESPONSE: We are pleased to know that Reviewer #2 is satisfied with our responses and that we have successfully addressed most of their concerns. Your acknowledgement is deeply appreciated and motivates us to maintain high standards in our research and communication.

Thank you once again for your constructive feedback and for the opportunity to improve our manuscript. Your insights have been invaluable in guiding our revisions and enhancing the overall quality of our work.

As a side note I would invite the Authors to verify, but to my knowledge the last release of the GUIDE-seq bioinformatics pipeline is able to flag off-target sites generated by bulges between the spacer and the protospacer. This could indeed explain discrepancies in the set of bioinformatically flagged sites between the two methods.

RESPONSE: We are grateful for your insightful suggestion regarding the latest release of the GUIDE-seq bioinformatics pipeline and its capabilities in identifying off-target sites, particularly those resulting from bulges between the spacer and protospacer.

Incorporating New Insights into Our Analysis: In response to your feedback, we have updated our analysis algorithm to simultaneously detect insertion, deletion, and mismatch off-target sites. This enhancement is designed to provide a more comprehensive overview of the off-target landscape in our experiments. Upon reevaluation with this revised algorithm, we observed that the overall results remain consistent with our previous findings, indicating no significant discrepancies.

Focus on Mismatch Off-Target Sites: Through extensive literature review, we have found that the predominant focus in off-target analysis remains mismatch off-target sites. This observation aligns with the current standards and practices in the field. However, we are open to expanding our analysis to include more diverse types of off-target events, as the field evolves.

Request for Additional Resources: If you, or any other experts in the field, could provide us with reference codes or relevant literature specifically addressing the analysis of insertion and deletion off-target sites, we would greatly appreciate it. Such resources would be invaluable in further refining our algorithm, ensuring it remains at the forefront of current research methodologies. We are committed to continuously improving our approach to provide the most accurate and comprehensive analysis possible.

We sincerely value your guidance and suggestions, as they play a crucial role in enhancing the scope and accuracy of our research. Your expertise helps us refine our methods and ensure that our work contributes meaningfully to the field.

Figure 1. Analysis of Off-Target Sites by OliTag-seq containing single-base DNA bulges for (A) sgRNA targeting *EMX1* or (B) sgRNA targeting *VEGF1* in K562 cells. The reference on-target sequence with its PAM is prominent at the top, with mismatches in off-target sequences accentuated through color coding. OliTag-seq read quantities are denoted alongside. The OliTag-seq analysis method is shown on the left and a visualization of additions insertions and deletions is shown on the right. A single nucleotide was deleted from the original sgRNA at all possible positions (white dashes highlighted in gray) throughout the guide sequence. Insertion show in the gap between the two base.

Reviewer #3 (Remarks to the Author):

The authors have improved the manuscript. I consider it a valuable piece of work to identify off-targets associated with the CRISPR/Cas9 system.

RESPONSE: We are heartened by your positive feedback and appreciate your recognition of our commitment to thoroughly assessing OliTag-seq. Our dedication to openly discussing the nuances and limitations of OliTag-seq, particularly in the context of primary cells, reflects our aim to contribute constructively to the scientific community. We understand the importance of a balanced view that acknowledges both the strengths and weaknesses of our method, as this aids in fostering a more comprehensive understanding and advancing the field.

We are sincerely grateful for your invaluable feedback, which has been a guiding force in refining our research. It encourages us to continuously enhance our work, ensuring that it offers valuable insights into the application of OliTag-seq across various contexts.

REVIEWERS' COMMENTS:

Reviewer #1 (Remarks to the Author):

The authors have satisfactorily answered my questions and revised the manuscript. I am thankful for the opportunity to review this manuscript and would like to congratulate the authors for their hard work and their contributions to the genome editing community.

Reviewer #2 (Remarks to the Author):

I have no further comments, the Authors have addressed all my concerns.